# Superposed Decoding: Multiple Generations from a Single Autoregressive Inference Pass

**Ethan Shen**◇  **Alan Fan**◇  **Sarah Pratt**◇  **Jae Sung Park**◇  **Matthew Wallingford**◇
**Sham Kakade**†  **Ari Holtzman**‡  **Ranjay Krishna**◇  **Ali Farhadi**◇  **Aditya Kusupati**◇∗

◇University of Washington  †Harvard University  ‡University of Chicago

{ethans03, kusupati}@cs.washington.edu

## Abstract

Many applications today provide users with multiple auto-complete drafts as they type, including GitHub's code completion, Gmail's smart compose, and Apple's messaging auto-suggestions. Under the hood, language models support this by running an autoregressive inference pass to provide a draft. Consequently, providing $k$ drafts to the user requires running an expensive language model $k$ times. To alleviate the computation cost of running $k$ inference passes, we propose Superposed Decoding, a new decoding algorithm that generates $k$ drafts at the computation cost of one autoregressive inference pass. We achieve this by feeding a superposition of the most recent token embeddings from the $k$ drafts as input to the next decoding step of the language model. At every inference step we combine the $k$ drafts with the top-$k$ tokens to get $k^2$ new drafts and cache the $k$ most likely options, using an n-gram interpolation with minimal compute overhead to filter out incoherent generations. Our experiments show that $k$ drafts from Superposed Decoding are at least as coherent and factual as Nucleus Sampling and Greedy Decoding respectively, while being at least $2.44\times$ faster for $k \geq 3$. In a compute-normalized setting, user evaluations demonstrably favor text generated by Superposed Decoding over Nucleus Sampling. Superposed Decoding can also be combined with other decoding strategies, resulting in universal coverage gains when scaling inference time compute. Code and more examples open-sourced at https://github.com/RAIVNLab/SuperposedDecoding.

## 1 Introduction

Commercial surveys find that 80% of e-commerce websites provide autocomplete as a feature [21]. With the proliferation of large language models, autocomplete drafts are now ubiquitous in an even wider range of applications. Examples include short draft suggestions in Gmail Smart Compose [8] and code snippets in GitHub Copilot [7]. These options provide users with the ability to consider different phrasings and increase the likelihood of having at least one suggestion that reflects their intent. While language models (LMs) [34] now power these multiple suggestions, each additional suggestion necessitates another inference pass (batch size = 1), making it computationally expensive.

Language models use autoregressive inference to generate a plausible sequence of next tokens for a given prefix [40]. The next token generated depends on the prefix and the previously generated tokens. Decoding algorithms like Greedy Decoding, Top-$k$ Sampling [12], Beam Search, and Nucleus Sampling [18] present various ways of obtaining generations during autoregressive inference. For a prefix, Greedy (maximum likelihood) Decoding picks the most probable token at every autoregressive timestep, eventually generating only one auto-completion suggestion. Instead of making a greedy

---

∗AK is currently at Google DeepMind

38th Conference on Neural Information Processing Systems (NeurIPS 2024).

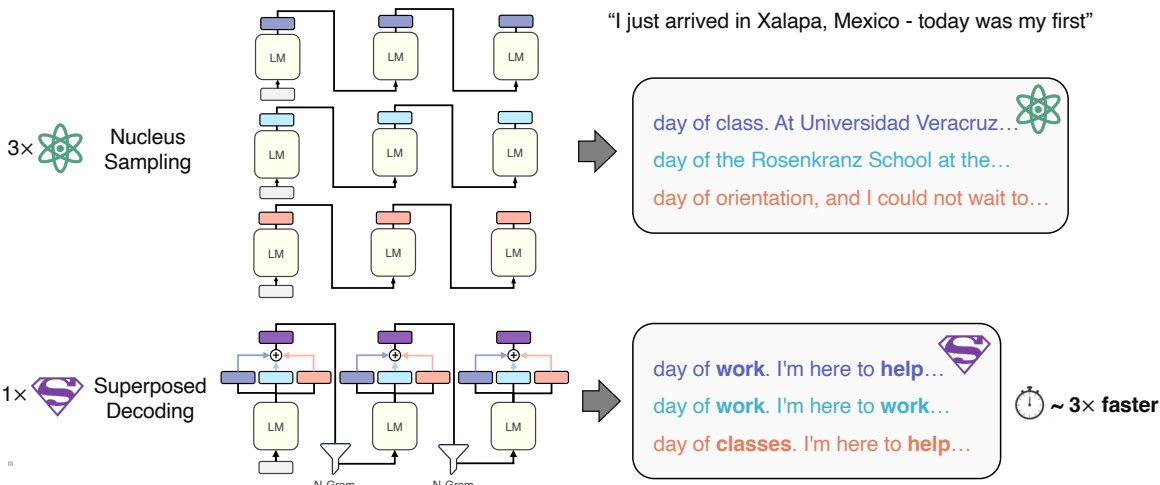

Figure 1: To generate multiple $k$ auto-complete suggestions for a prefix using an LM, the existing decoding methods like Nucleus Sampling need $k$ inference passes. In contrast, Superposed Decoding can generate $k$ suggestions at the cost of a single inference pass while being as coherent and factual.

choice at every timestep, we can also sample from the top-$k$ most probable tokens to generate the sequence [12]. Also leveraging the top-$k$ most probable tokens is Beam Search, which picks the most probable $k$ beams from the $k^2$ possible drafts at every timestep. Alternatively, Nucleus Sampling [18], particularly effective at generating natural-sounding text, grows generations by sampling from the collection of the smallest subset of tokens that form a preset probability mass ($p$) at every timestep. Top-$k$ Sampling, Beam Search, and Nucleus Sampling offer the benefit of generating multiple suggestions. However, this comes at the cost of requiring multiple autoregressive inference passes.

We introduce Superposed Decoding (SPD) (Figure 1), a decoding algorithm that can generate mutiple ($k$) high-quality short generations using only a *single* autoregressive inference pass. At each autoregressive timestep during inference, Superposed Decoding feeds in the superposition (weighted combination) of the embeddings corresponding to the $k$ most recent drafted tokens (Section 3.1). After selecting the top-$k$ output tokens, SPD expands the existing $k$ drafts with them, resulting in $k^2$ potential new drafts. Each draft has a probability score assigned to it, which is further smoothed using a combination of various $n$-gram models ($n \in [2, 6]$) [38, 25]. N-gram interpolation for smoothing helps improve coherency by selecting the most probable and coherent continuations (top-$k$ drafts) for the next autoregressive step (Section 3.2). The n-gram interpolation is computationally inexpensive and allows flexibility to change domains of interest during inference (eg., programming, healthcare, finance, etc.). Superposed Decoding's effectiveness can be attributed to the apparent linearity of representations in language models [35, 23] which we concurrently discovered (Section 3.3).

Our experiments demonstrate that Superposed Decoding on Llama-2-7B [41] can generate $k \geq 1$ drafts that are as coherent, in terms of perplexity, as Nucleus Sampling (Section 4.1). However, SPD achieves this with a single inference pass, making it at least $2.44\times$ faster than any other standard decoding methods for $k \geq 3$ (Section 4.3). The ability to generate multiple drafts at the cost of one also increases the coverage significantly for fact-based evaluation tasks like TriviaQA and Natural questions – where SPD increases the chance of generating the correct answer by at least $5\%$ when using 3 drafts (Section 4.2). Through an extensive human evaluation for a wide range of prefixes, we show that SPD generations are as preferred as Nucleus Sampling in direct comparison (1v1) while outperforming by up to $20\%$ in a compute normalized setting (3v1 and 3v2) (Section 4.4). Finally, we find that combining Superposed Decoding and other decoding strategies (e.g. Nucleus Sampling) results in substantial accuracy improvements when scaling inference compute (Section 4.5).

Superposed Decoding can help improve user experience by offering significant computational efficiency while maintaining accuracy for various writing tasks that often benefit from multiple short draft suggestions. Additionally, Superposed Decoding is extremely generalizable, works across different language models like Mistral 7B, and is capable of generating long-form content reliably, all while being nearly as diverse in suggestion as Nucleus Sampling (Section 5).

## 2 Related Work

Decoding algorithms determine how tokens are selected from a language model's next token distribution. This is typically done greedily with Beam Search or Greedy Decoding, or stochastically with Nucleus Sampling [18] or Top-$k$ Sampling [12]. Greedy Decoding operates by selecting the token with the highest probability at each timestep. However, locally optimal decisions can be globally suboptimal. On the other extreme, an exhaustive exploration of all token combinations is intractable with time complexity $\Theta(|\mathcal{V}|^T)$, where $\mathcal{V}$ is the vocabulary size and $T$ the total number of timesteps. A practical compromise is Beam Search, which continually caches the top-$B$ most likely generations, where $B$ is the number of beams or drafts. Thus, Beam Search is guaranteed to find more likely generations than Greedy Decoding while avoiding its drawbacks.

Alternatively, Nucleus Sampling and Top-$k$ Sampling decode probabilistically. To avoid degeneration, Top-$k$ Sampling truncates the probability mass to the $k$ most probable tokens, whereas Nucleus Sampling truncates the probability mass to the smallest possible set of words whose cumulative probability exceeds probability $p$. Because of Nucleus Sampling's propensity to produce unique and unrepetitive outputs, it has become the standard [4, 22, 42], though Greedy Decoding is still favored for tasks such as short question answering [41].

Generating multiple drafts linearly scales the number of inference passes in these decoding methods. Multi-token prediction [14, 36] addresses this by pre-training an LM having $n$ independent softmax heads that predict $n$ future tokens in parallel. When using multi-token prediction with speculative decoding, exact inference is $3\times$ faster. In a similar vein, Medusa [5] reduces the number of decoding steps by adding extra decoding heads, using a tree-based attention mechanism to generate candidates while simultaneously verifying them in each decoding step. Through this, Medusa achieves a $2.2\times$ reduction in inference latency while maintaining generation quality. However, multi-token prediction requires re-training and Medusa requires additional fine-tuning for the extra decoding heads.

Superposed Decoding (SPD), on the other hand, can be easily integrated **out of the box** without any additional training on any language model. Further, Superposed Decoding is complementary to other decoding methods like Medusa and multi-token prediction, as well as efficiency techniques like speculative decoding [30, 26, 6], quantization [10, 32, 19], pruning [13, 39, 28], and architectural optimizations [37, 2, 1, 43, 27, 11].

## 3 Superposed Decoding (SPD)

Given a text input as a prefix, Superposed Decoding uses an autoregressive LM $f_\theta$ to produce $k$ viable completion drafts in one inference pass.

**First Timestep.** Let $x$ denote a generated token in the vocabulary $\mathcal{V}$ and $M = (x_1, \ldots, x_m)$ represent an initial prefix of $m$ tokens. Our goal is to generate $k$ unique drafts starting from the prefix. The next token distribution at the first timestep $m + 1$ is: $p_\theta(x_{m+1}|x_1, \ldots x_m) = p_\theta(x_{m+1}|M)$.

For the first timestep, each of the $k$ drafts is initialized as the prefix so we use the same next token distribution for all drafts. We grow draft $d_i$ by greedily selecting the $i^{\text{th}}$ most probable token, making:

$$d_i = (M, x_{m+1}^i)$$

where $x_t^i$ is the token at timestep $t$ for the $i^{\text{th}}$ draft. We also track each draft's probability $p_i$ as its score, which is initially the probability of its first token $p_\theta(x_{m+1}^i|M)$. Consequently, the set of drafts $D$ after the first timestep is:

$$D = \begin{pmatrix} (M, x_{m+1}^1) \\ \vdots \\ (M, x_{m+1}^k) \end{pmatrix} \text{ with probabilities } P = \begin{pmatrix} p_\theta(x_{m+1}^1|M) \\ \vdots \\ p_\theta(x_{m+1}^k|M) \end{pmatrix} \tag{1}$$

**Next Timesteps.** As the input to the LM at timestep $t$, we construct $\tilde{x}_{t-1}$, which is a superposition (weighted linear combination) of the embeddings for the most recent token $x_{t-1}$ of the $k$ drafts. This means that $\tilde{x}_{m+1}$, the input to the LM at the second timestep $m + 2$, is the superposition of the tokens $x_{m+1}^i$ for $i = 1, \ldots, k$. We use this *superposed embedding* as a single and accurate approximation

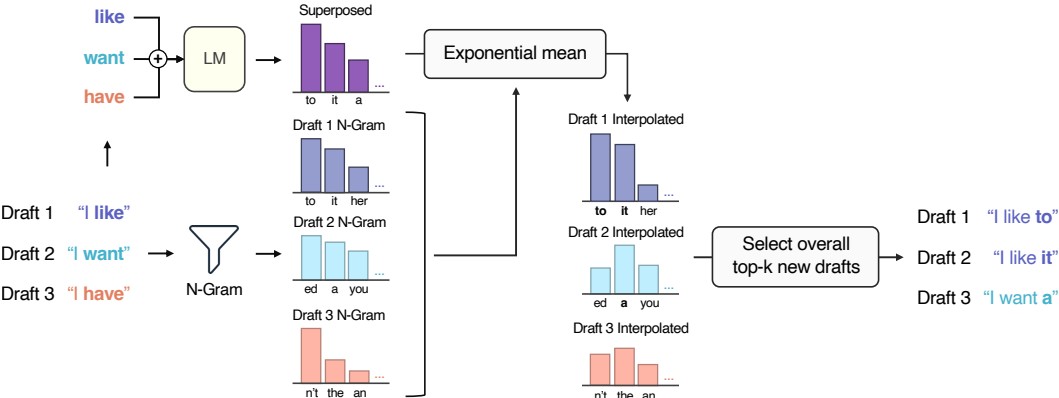

Figure 2: Superposed Decoding relies on feeding a superposed token embedding – based on the most recent tokens from the current $k$ drafts – as the input during the auto-regressive inference step. This generates $k^2$ new drafts using the existing $k$ drafts and the top-$k$ output tokens at the new timestep. Finally, keep the top-$k$ drafts after filtering with an n-gram interpolation to improve coherency.

for the most recent token across all $k$ drafts at the $(t-1)^{\text{th}}$ timestep. We run autoregressive inference over $\tilde{x}_t$ once for all $k$ drafts instead of the usual once per draft, allowing us to formulate inference as:

$$p_\theta(x_t|M, \tilde{x}_{m+1}, \dots, \tilde{x}_{t-1})$$

Because each draft contains its own contextual clues and syntax, we cannot blindly use the distribution of $x_t$ for each draft. Instead, we interpolate the superposed distribution $p_\theta(x_t|M, \tilde{x}_{m+1:t-1})$ with a draft-specific next token distribution from an n-gram model to get a final distribution $p_f(x_t^i|M, x_{m+1:t-1})$ that is unique to each draft. Next, we rank the draft options by the joint probability of their respective previous draft $p_i$ and their selected token. We choose the top $k$ options as drafts for the next timestep and update their probabilities. This gives:

$$D = \begin{pmatrix} (M, x_{m+1}^1, \dots, x_t^1) \\ \vdots \\ (M, x_{m+1}^k, \dots, x_t^k) \end{pmatrix} \text{ with probabilities } P = \begin{pmatrix} p_\theta(x_{m+1}^1|M) \prod_{s=m+2}^t p_f(x_s^1|M, x_{m+1:s-1}^1) \\ \vdots \\ p_\theta(x_{m+1}^k|M) \prod_{s=m+2}^t p_f(x_s^k|M, x_{m+1:s-1}^k) \end{pmatrix}$$
$$(2)$$

We continue generation until the maximum generation length or stop token is reached. In the following sections, we break down the process in detail. We also present pseudocode in Appendix A.

### 3.1 Token Superposition

During training, language models learn a representation (token embedding) $z \in \mathcal{R}^d$ for every token $x \in \mathcal{V}$. We leverage this representation at timestep $t$ to construct $\tilde{x}_t$, weighing the representation for $x_{t-1}^i$ by a coefficient $\gamma_i$.

$$\tilde{x}_t = \sum_{i=1}^k \gamma_i \cdot z_{t-1}^i \text{ where } \sum_{i=1}^k \gamma_i = 1 \tag{3}$$

The performance of $\tilde{x}_t$ is highly dependent on choosing the appropriate weight for each embedding $z_{t-1}^i$. We find that the best strategy is to set $\gamma_i$ to the normalized probability of draft $i$ such that

$$\gamma_i = \frac{p_i}{\sum_{j=1}^k p_j} \tag{4}$$

where $p_i$ is the probability of the $i^{\text{th}}$ draft, introduced in Section 3. This allows us to directly tie the weight of a token to the likelihood that it will be preserved in future timesteps, reducing drift between the superposed embeddings and the drafts they represent.

### 3.2 N-Gram Interpolation

We construct each draft's n-gram distribution $p_{\text{ngram}}(x_t^i|M, x_{m+1}^i, \dots, x_{t-1}^i)$ by interpolatively smoothing the next token distributions over a set of $n$-gram models using weights $\lambda$, where $n \in [2, 6]$.

$$p_{\text{ngram}}(x_t^i|M, x_{m+1}^i, \dots, x_{t-1}^i) = \sum_{n=2}^{6} \lambda_n \cdot p_{n\text{-gram}}(x_t^i|M, x_{m+1}^i, \dots, x_{t-1}^i) \tag{5}$$

We base our interpolation weights on weights for RedPajama found by Liu et al. [31], with additional tuning (specifics in Appendix B). However, domain-specific n-gram models can easily be swapped in for specific tasks such as code generation [20]. We use the exponential mean of $p_\theta$ and $p_{\text{ngram}}$ as our final distribution $p_f$, where $\alpha$ is a hyperparameter controlling the impact of the n-gram distribution:

$$p_f(x_t^i|M, x_{m+1:t-1}^i) = p_\theta(x_t|M, \tilde{x}_{m+1:t-1})^{1-\alpha} \cdot p_{\text{ngram}}(x_t^i|M, x_{m+1:t-1}^i)^{\alpha} \tag{6}$$

This means that when generating, we only consider tokens appearing in both the Superposed Decoding and n-gram distributions. If there is no overlap between the distributions, then we instead calculate

$$p_f(x_t^i|M, x_{m+1:t-1}^i) = \delta \cdot p_\theta(x_t|M, \tilde{x}_{m+1:t-1})^{1-\alpha} \tag{7}$$

without interpolation, where $\delta$ is a penalty term that disincentivizes selecting an uninterpolated draft for the next timestep. This approach is the backbone of Superposed Decoding (Equation 2) and allows us to create context-aware next-token distributions for *each* draft with only *one* inference pass.

### 3.3 Superposed Decoding Semantically Approximates Beam Search

Superposed Decoding relies on the ability of the underlying model to preserve the linear relationship between $\tilde{x}_t$ and its component vectors $z_{t-1}^i$ [23]. More formally, if $\tilde{x}_t$ is the input to the LM $f_\theta$, then $f_\theta(\tilde{x}_t) = \sum_{i=1}^{k} \gamma_i \cdot f_\theta(z_{t-1}^i)$ should also be true ($\gamma_i$ defaults to draft probability in SPD). As long as this holds, the combination of a superposed embedding and n-gram smoothing should allow us to generate completions that resemble those from methods such as Beam Search.

We test this linearity by computing the cosine similarity between a set of superposed embeddings $\{\tilde{x}\}$ and the linear combination of their component embeddings across 20 timesteps on Llama-2-7B. At each timestep, we first use Beam Search to generate tokens for three beams. We then manually input the superposed embedding of the three tokens into a model using Superposed Decoding . Finally, we measure the alignment between $f_\theta(\tilde{x}_t)$ and $\sum_{i=1}^{k} \gamma_i \cdot f_\theta(z_{t-1}^i)$ using cosine similarity $\cos(a, b)$ as:

$$\cos(f_\theta(\tilde{x}_t), \sum_{i=1}^{k} \gamma_i \cdot f_\theta(z_{t-1}^i)) \tag{8}$$

We compute the cosine similarities for ten randomly sampled batches of ten prefixes each from the OpenWebText training split and plot the mean cosine similarities across batches, as well as the standard deviation (Figure 3). We find that Llama-2-7B is sufficiently linear up to 10 timesteps across all layers. However, this linearity is imperfect, and Superposed Decoding and Beam Search eventually diverge over time. Owing to this, we identify 10 timesteps as the optimal generation length. We also show how linearity changes through the layers within each timestep in Appendix H.

## 4 Experiments

We evaluate Superposed Decoding by analyzing generation quality, factuality, and latency. We demonstrate that Superposed Decoding improves over Nucleus Sampling and Greedy Decoding

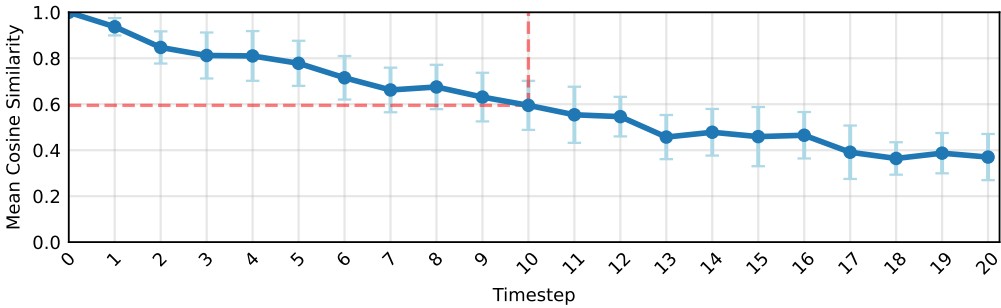

Figure 3: Llama-2-7B maintains the linear relationship between superposed embeddings and the component token embeddings, with mean cosine similarity $\geq 0.6$ for the first 10 timesteps.

| | OpenWebText | Over a century ago, the RMS Titanic's fate | Wayne said: I like the calculations, but like all theoretical | Hello You Jump-Junkies! Apologies for |
|---|---|---|---|---|
| Nucleus | | was sealed when it struck an iceberg on | calculations, it is best to try it out in | the long silence, but we've been busy |
| Superposed | | was sealed when it **struck** an iceberg **and** | **calculations**, they are only as good as the **data** | the long **wait**. I've been busy with |
| | | was sealed when it **hit** an iceberg **and** | **models**, they are only as good as the **data** | the long **silence.** I've been busy with |
| | | was sealed when it **hit** an iceberg **on** | **calculations**, they are only as good as the **assumptions** | the long **silence,** I've been busy with |

Figure 4: Qualitative text generations in a compute-normalized setting for Superposed Decoding and Nucleus Sampling with prefixes sampled from OpenWebText. See Appendix G.1 for more.

in generation perplexity (Section 4.1), fact-based benchmarks (Section 4.2), and wall-clock time (Section 4.3). We also conduct a user study highlighting that users prefer Superposed Decoding over Nucleus Sampling in a compute-normalized setting (Section 4.4). Finally, we show that Superposed Decoding improves performance while scaling inference compute (Section 4.5). We include example generations of Superposed Decoding in Figure 4, with more in Appendix G.1.

We implement Superposed Decoding on the base version of Llama-2-7B [42] for the majority of our experiments, running on one A40 GPU. We do not change model weights. For perplexity evaluations, we use Llama-2-70B on eight A40 GPUs. We assume a batch size of one for all experiments.

For n-gram interpolation, we construct n-gram models using 200,000 documents (roughly 200 million tokens) randomly sampled from the RedPajama dataset, an open-source replication of Llama-2's training dataset [9]. We represent each n-gram model internally as a frequency table, storing each unique n-gram and its corresponding count. This enables faster lookup and is the basis behind the compute reduction that we offer. Our n-gram models require approximately 14 GB of disk storage overall, split 57 MB, 433 MB, 2.15 GB, 4.7 GB, and 6.8 GB for $n = 2$ to 6. While we interpolate up to $n = 6$ in this work, we note that in practice the benefits of n-gram interpolation saturate past $n = 4$, suggesting that the number of n-gram models can be lowered to reduce storage (Appendix C).

## 4.1 Generation Quality

We test generation quality on OpenWebText's test split [15], which consists of 5,000 web-scraped documents. For each document, we use the first 15 tokens as the prefix and generate $k = 3$ drafts of 10 tokens with a batch size of 1. We decide to focus on short generations because drafts are typically a short-form task. Before running experiments, we identify the optimal interpolation weight $\alpha$ and temperature $\tau$ by optimizing for the lowest average perplexity across three drafts on the validation split. We list the specific hyperparameter values that we use in Appendix B.

We only evaluate one draft of the baselines in order to match the compute used by Superposed Decoding. We find that while Nucleus Sampling and Greedy Decoding outperform Superposed Decoding on a per-draft basis, the average best perplexity from Superposed Decoding is **5% lower** than that of Nucleus Sampling. From the point of view of a user, this means for each prefix, at least one draft can be expected to be on par with Nucleus Sampling and Greedy Decoding. The other drafts all come free of additional compute. In the following Sections 4.2 and 4.4, we show that this diversity is beneficial for both factuality and human preference.

Table 1: Superposed Decoding is natural-sounding and has lower average perplexity than Nucleus Sampling, which typically approximates human writing.

| | Nucleus | Beam/Greedy | N-Gram | Superposed Decoding | | | |
|---|---|---|---|---|---|---|---|
| Draft # | - | - | - | 1 | 2 | 3 | Best |
| Avg Perplexity | 5.17 | 3.77 | 152.75 | 5.03 | 7.97 | 10.05 | 4.63 |

## 4.2 Fact-Based Evaluation

Next, we test the ability of Superposed Decoding to generate not only coherent but also accurate completions. We assess this using exact match precision (P@$k$) for $k = 1, 2, 3$ on TriviaQA [24] and Natural Questions [29], two common benchmarks for short answer generations. We decide not to

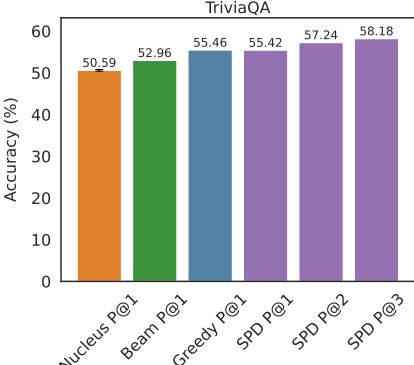
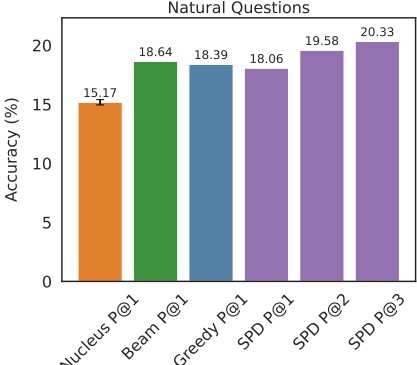

Figure 5: Superposed Decoding is as accurate as Greedy Decoding for P@1 and increases the fact-based coverage using multiple drafts (P@2,3) on TriviaQA (**left**) and Natural Questions (**right**).

use multiple choice datasets such as MMLU [17] and OpenBookQA [33] because multiple choice questions are trivial when using multiple drafts, unfairly advantaging Superposed Decoding.

In Figure 5, we show that Superposed Decoding outperforms Nucleus Sampling and Beam Search at P@1 in a zero-shot setting, with three drafts providing accuracy gains of up to $2.72\%$ in TriviaQA and $1.69\%$ in Natural Questions. These results demonstrate that Superposed Decoding substantially increases the likelihood of getting a factually correct generation in addition to one that is coherent.

## 4.3 Latency

Superposed Decoding presents a significant *theoretical* reduction in latency, but it is important to investigate how well this translates to the real-world. In Figure 6, we show that dictionary lookups are the only additional cost incurred by Superposed Decoding, barring which Superposed Decoding has near-constant compute cost. Even so, the total cost of Superposed Decoding is significantly lower than other decoding methods, with Superposed Decoding $\mathbf{2.44\times}$ faster on three drafts and $\mathbf{3.54\times}$ faster on eight compared to Nucleus Sampling, the next fastest.

It is important to note that Superposed Decoding results are obtained using unoptimized code. Our n-gram models are implemented using single-threaded lookup on Python dictionaries, and we do not cache any lookup results. This has an enormous, visible impact. In addition, Liu et al. [31] propose the use of suffix arrays for n-gram models, allowing a near-instantaneous lookup of arbitrary-length n-grams in trillion-token corpora. These techniques open up multiple avenues for additional speedup.

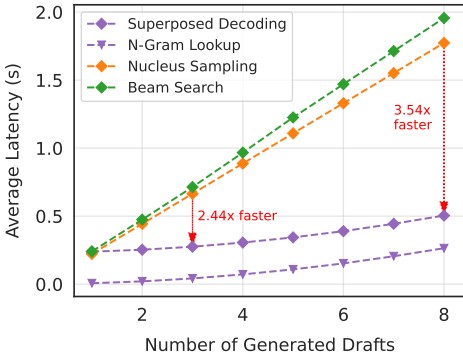
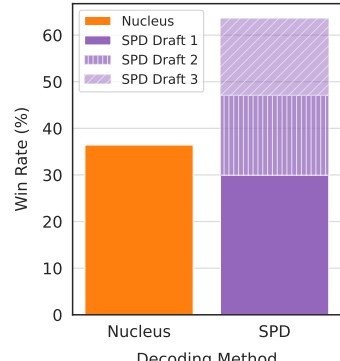

Figure 6: Average wall clock time to generate 10 token long drafts, with batch size $= 1$, from a 15 token prefix on OpenWebText. Superposed Decoding is significantly faster for all $k > 1$, with n-gram lookup costs being a major factor for $k \geq 4$, which can be optimized further.

Figure 7: Drafts are ordered by the probability they obtain during generation using SPD, which wins over Nucleus Sampling in a compute-normalized setting.

## 4.4 Human Evaluation

While perplexity and factuality are good proxies for textual quality, real human evaluation remains the best measure of coherency. Indeed, perplexity is an imperfect evaluator of long-form language modelling [44] while factuality metrics are inherently biased towards Greedy Decoding methods [41]. Consequently, to verify our findings, we conduct a random study asking human respondents to rank Superposed Decoding and Nucleus Sampling generations based on how well they complete a provided prefix. We compare against Nucleus Sampling because n-gram distributions from Nucleus Sampling are demonstrably the closest to those of humans [18].

**Setup.** We conduct our study using Amazon Mechanical Turk [16]. First, we randomly sample 1,000 prefixes from the OpenWebText test set, truncating to the first 15 tokens as in Section 4.1. Next, we manually remove prefixes that are grammatically incorrect, such as text from website toolbars. From the remaining prefixes, we generate three Superposed Decoding drafts and one Nucleus Sampling draft in a compute-normalized setting, randomizing their order. Finally, we filter out prefixes with duplicate or unparseable generations (e.g. emojis). After preprocessing, 707 prefixes are left.

It is important to note that one Nucleus Sampling generation is $20\%$ less expensive than three Superposed Decoding drafts, shown in Section 4.3. However, two Nucleus Sampling drafts disadvantages three Superposed Decoding drafts by $60\%$. Therefore, we decide to run our main survey using the first setup, which is closest to equal compute, but also conduct smaller surveys in 2v3 and 1v1 settings.

**Results.** We define a Superposed Decoding "win" as when one of the Superposed Decoding drafts is ranked first. As shown in Figure 7, we find that Superposed Decoding generations are preferred approximately $\mathbf{63.6}\%$ of the time . If every Superposed Decoding draft was consistently the same quality as Nucleus Sampling, then we would expect the rankings to be uniform, resulting in a $75\%$ win rate. However, this is not the case, suggesting that Nucleus Sampling is more reliable than any *individual* draft, but the *aggregate* of drafts provides a diversity that is highly valuable to users.

We run the two smaller-scale iterations of the study with 100 prefixes each. In the 2v3 setting, we ask users to rank two nucleus drafts and three superposed drafts to investigate whether the benefits of Superposed Decoding persist even when compute favors Nucleus Sampling. In the 1v1 setting, users choose between one nucleus draft and the lowest perplexity superposed draft, removing any potential bias caused by unequal numbers of drafts. As shown in Figure 12, in both situations, users still prefer Superposed Decoding over Nucleus Sampling $\mathbf{60.6}\%$ and $\mathbf{51.4}\%$ of the time respectively. While the surveys have higher variance due to their small size, they cement Superposed Decoding as a strong alternative to Nucleus Sampling. We show details on the additional studies in Appendix F.1 and present our survey page in Figure 13.

## 4.5 Inference-Time Scaling

Superposed Decoding also provides significant benefits for inference time compute scaling. Superposed Decoding is completely complimentary to other decoding methods, expanding semantic coverage at no extra compute. For instance, if Nucleus Sampling is used to generate $n$ drafts, Superposed Decoding with $k$ drafts can be spliced in at any timestep to strategically produce $nk$ drafts

Table 2: Coverage on TriviaQA and Natural Questions in a normalized compute setting comparing vanilla Nucleus Sampling ($NS$) to the combination of Nucleus Sampling and Superposed Decoding ($NS_{SPDk}$), where $k$ is the number of Superposed Decoding drafts generated on top of each Nucleus Sampled generation. $NS_{SPD}$ results in better coverage at nearly every compute budget $n$.

| Task | Decoding Method | Compute ($n$) | | | | | | | | | | |
|---|---|---|---|---|---|---|---|---|---|---|---|---|
| | | 1 | 10 | 20 | 30 | 40 | 50 | 60 | 70 | 80 | 90 | 100 |
| TriviaQA | $NS$ | 51.04 | 68.75 | 70.31 | 71.87 | 72.92 | 74.48 | 74.74 | 75.26 | 75.78 | 76.30 | 76.56 |
| | $NS_{SPD2}$ | 51.30 | 68.75 | 70.57 | 72.66 | 74.74 | 75.78 | 76.30 | 76.82 | 78.39 | 79.17 | 79.43 |
| | $NS_{SPD3}$ | **51.82** | **70.57** | **74.22** | **75.52** | **77.34** | **77.87** | **78.39** | **78.65** | **79.17** | **79.43** | **79.95** |
| Natural Questions | $NS$ | 14.32 | **32.55** | **36.98** | 38.54 | 40.36 | 41.15 | 41.67 | 41.93 | 42.19 | 42.71 | 42.97 |
| | $NS_{SPD2}$ | 15.36 | 31.25 | 34.90 | 38.02 | 39.84 | 41.41 | 41.67 | 42.45 | 43.75 | 43.75 | 43.75 |
| | $NS_{SPD3}$ | **15.63** | 31.25 | **36.98** | **39.06** | **41.15** | **42.71** | **43.75** | **43.75** | **44.27** | **44.79** | **45.57** |

at no extra cost, using the Nucleus Sampled generations as prefixes. This bolsters every Nucleus Sampling generation with $k$ local searches (example in Appendix G.2).

This property is valuable for repeated sampling [3], a technique where the number of generations is scaled at inference time to increase coverage – the proportion of questions that can be answered correctly using at least one of the generations. While repeated sampling typically uses Nucleus Sampling, combining Superposed Decoding and Nucleus Sampling ($NS_{SPD}$) produces even larger coverage gains. In Table 2, we compare the coverage of vanilla Nucleus Sampling and $NS_{SPD}$ in an equal compute setting up to 100 Nucleus Sampling drafts. We find that $NS_{SPD}$ results in higher coverage *across the board* on both TriviaQA [24] and Natural Questions [29], highlighting Superposed Decoding as a powerful method to increase the impact of scaling inference compute.

## 5    Further Analysis and Ablations

### 5.1    Results on Mistral 7B

We also implement Superposed Decoding on pre-trained Mistral 7B [22] and conduct the same experiment as Section 4.1 with one change. In Section 4.1, it was possible to evaluate the perplexity of the 10 generated tokens exactly because the generating model (Llama-2-7B) and the evaluation model (Llama-2-70B) use the same tokenization. This is not the case for Mistral 7B and Llama-2-70B. Consequently, we calculate perplexity for Mistral 7B over all tokens but the first five, ensuring that the entire generation is included. While this approach also includes several tokens from the initial prefix in perplexity calculations, they are redundant across generations, thus preserving relative ordering.

Table 3 compares the resulting perplexities. Like with Llama-2-7B, the average best draft perplexity using Superposed Decoding is lower than that of Nucleus Sampling, demonstrating that Superposed Decoding is adaptable to other language models out of the box.

### 5.2    Textual Analysis

Next, we extensively investigate the diversity and repetition of Superposed Decoding in order to better understand its properties.

**Repetition within Generations.** Repetition is a well-documented issue in all decoding methods but is most prevalent when decoding greedily [18]. We explore to what extent, if any, Superposed Decoding degenerates as well. To measure repetition, we calculate the proportion of unique unigrams, bigrams, and trigrams in each generation. The lower the uniqueness, the higher the repetition. In Figure 8, we plot results for Superposed Decoding and Nucleus Sampling for several generation lengths. Because drafting is usually a short-form task, we only consider generation lengths up to 20 tokens. We find that Superposed Decoding does not repeat significantly more than Nucleus Sampling in this range, suggesting that degeneration is not an issue in most use cases. This is especially true for bigrams and trigrams, which are more reliable indicators of degeneration than unigrams. However, we qualitatively observe that repetitions become frequent after 100 tokens. To address this, we propose Superposed Decoding Resets, which we explain in Section 5.3.

**Diversity across Drafts.** To measure diversity, we apply Self-BLEU [45] on drafts and then calculate the average across prefixes. We compute Self-BLEU by calculating the BLEU score of each draft with the other $k - 1$ drafts as references. Hence, a low Self-BLEU signifies high diversity, while a high Self-BLEU suggests low diversity. After calculating Self-BLEU for varying prefix lengths, generation lengths, and numbers of drafts, we find that generation length is the most impactful hyperparameter for diverse drafts. We plot Self-BLEU against generation length in Figure 8, demonstrating that shorter generations significantly increase diversity.

Table 3: Superposed Decoding generalizes across language models like Mistral 7B as shown here with similar results on coherency, as Llama-2-7B, when evaluated using Llama-2-70B.

|  | Nucleus | Superposed Decoding | | | |
| --- | --- | --- | --- | --- | --- |
| Draft # | - | 1 | 2 | 3 | Best |
| Avg Perplexity | 11.42 | 11.34 | 12.74 | 13.63 | 10.87 |

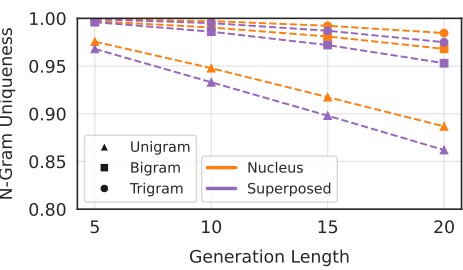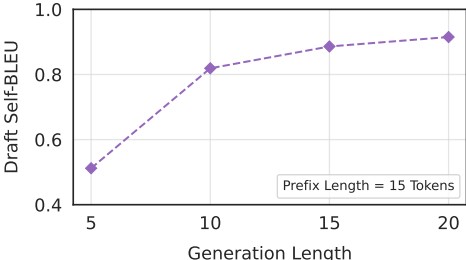

Figure 8: **Left:** Minimal difference in repetition for Superposed Decoding and Nucleus Sampling in short generations. **Right:** Generation length is an effective knob for adjusting the diversity of drafts. Both experiments use a prefix length of 15 tokens.

### 5.3 Superposed Decoding Resets

To reduce long-form repetition in Superposed Decoding, we propose resetting superposed tokens every $s$ timesteps, where $s$ is a user-selected hyperparameter. On each reset step we sample one of the $k$ drafts and restart draft generation. Resetting helps Superposed Decoding escape repetitive loops while reestablishing linearity, which Figure 3 shows to deteriorate over time. This is similar to a user selecting one of the $k$ drafts while typing. We find that resetting noticeably improves long-form generation quality (examples in Appendix I) and leave further investigation for future work.

Further, we also ablate on prefix length, generation length, and number of drafts. Figures 9, 10, and 11 in Appendix E highlight that Superposed Decoding is robust to all three hyperparameters, with lower perplexity than Nucleus Sampling in nearly all settings tested.

## 6 Discussion and Conclusion

While we demonstrate that Superposed Decoding is an effective technique for multiple draft generation, Superposed Decoding is limited by the quality of the n-gram models used, which are essential for maintaining coherence. In addition, while Superposed Decoding drafts are *syntactically* diverse, they are not often *semantically* diverse. We suspect that the orthogonality of token embeddings discovered by Jiang et al. [23] is a potential solution. While our initial experiments did not show diversity gains, we believe that orthogonality is promising and is a logical next step for future work. We also note that mechanisms, like batching, that increase throughput of decoding algorithms are complementary to Superposed Decoding.

In conclusion, we present **Superposed Decoding**, a novel decoding method that superposes token embeddings to generate multiple short generations at the cost of one. We demonstrate that Superposed Decoding improves on Nucleus Sampling in terms of generation quality and human preference. The plurality of choices from Superposed Decoding also leads to better performance on common benchmarks and expands coverage at scale. We envision that the latency reduction from Superposed Decoding will make it practical to apply large pre-trained language models on drafting tasks where compute is often a barrier for deployability. Finally, Superposed Decoding can be deployed in messaging applications using n-grams personalized to each user, where the number of n-grams can be reduced to save storage without compromising performance.

## Acknowledgments

We are grateful to Jiacheng (Gary) Liu for his assistance in using infinigram for initial experimentation. We also thank Alisa Liu, Vivek Ramanujan, Reza Salehi, Prateek Jain, Yashas Samaga B L, and Yejin Choi for their feedback. We also acknowledge the computing resources and support from HYAK at the University of Washington, FAS RC at Harvard University & Kempner Institute, Gradient AI and research GCP credit award from Google Cloud and Google Research. Ali Farhadi acknowledges funding from the NSF awards IIS 1652052, IIS 17303166, DARPA N66001-19-2-4031, DARPA W911NF-15-1-0543, and gifts from Allen Institute for Artificial Intelligence and Google. Sham Kakade acknowledges funding from the Office of Naval Research under award N00014-22-1-2377. This work has been made possible in part by a gift from the Chan Zuckerberg Initiative Foundation to establish the Kempner Institute for the Study of Natural and Artificial Intelligence.

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

## A   Superposed Decoding Pseudo-Code

---

**Algorithm 1** Superposed Decoding algorithm generating $k$ drafts.

---

1: **Input:** Prefix $M = (x_1, \ldots, x_m)$, generation length $G$
2: Initialize $t \leftarrow m + 1$
3: Initialize $D \leftarrow \{M\}_k$              ▷ Draft store
4: **while** $t \neq m + G$ **do**
5:      **if** $t = m + 1$ **then**           ▷ First timestep
6:          $p_\theta \leftarrow f_\theta(M)$
7:          **for** $i = 1$ to $k$ **do**
8:              $x_{m+1}^i \leftarrow \text{Largest}(p_\theta, i)$      ▷ Take the $i^{\text{th}}$ most probable token
9:              $D_i \leftarrow \{D_i \cup x_{m+1}^i\}$
10:          **end for**
11:          $\tilde{x}_{m+1} \leftarrow \text{Superpose}(x_{m+1}^1, \ldots, x_{m+1}^k)$      ▷ Superpose the selected tokens
12:      **else**
13:          $p_\theta \leftarrow f_\theta(M, \tilde{x}_{m+1:t-1})$
14:          $p_\theta \leftarrow \text{KeepTopK}(p_\theta)$      ▷ Keep only the top $k$ tokens
15:          **for** $i = 1$ to $k$ **do**
16:              $p_{\text{ngram}} \leftarrow \text{N-Gram}(D_i)$      ▷ Calculate the n-gram distribution of draft $i$
17:              $p_f \leftarrow \text{Interpolate}(p_\theta, p_{\text{ngram}})$
18:              **for** $x_t^i \in p_f$ **do**
19:                  $D_{tmp} \leftarrow \{D_i \cup x_t^i\}$      ▷ Create new draft option
20:                  $D \leftarrow \{D \cup D_{tmp}\}$      ▷ Add draft option to draft store
21:              **end for**
22:          **end for**
23:          $D \leftarrow \text{TopK}(D)$      ▷ Keep the top $k$ drafts
24:          $\tilde{x}_t \leftarrow \text{Superpose}(x_t^1, \ldots, x_t^k)$      ▷ Superpose most recent tokens from drafts
25:      **end if**
26:      $t \leftarrow t + 1$
27: **end while**
28: **return** $D$

---

## B   Hyperparameter Choices

$\alpha$ **and** $\tau$**.**   We find that the best performing hyperparameters for coherent generation on Llama-2-7B are $\alpha = 0.54$ and $\tau = 0.06$, which we use in Section 4.1, 4.3, and 4.4. The only exception is in Section 4.2, where we disable n-gram smoothing.

**N-Gram Interpolation.**   The weights we use for interpolation are $[0.01, 0.04, 0.15, 0.18, 0.12]$ for $n = 2$ to $n = 6$.

## C   Perplexity Evaluation with Fewer N-Grams

We evaluate perplexity for Superposed Decoding on the OpenWebText test split by interpolating only $n$-grams from $n = 2$ to $n = 4$, requiring only 2.64 GB of storage. We use the same n-gram interpolation weights as in Section 3 but retune $\alpha$ and $\tau$ to 0.55 and 0.1 respectively. In Table 4, we show that the perplexity of the average best generation is almost identical to the average Nucleus Sampling perplexity. This demonstrates that Superposed Decoding can be effectively implemented with minimal storage overhead in storage-constrained devices like smartphones.

## D   Standard Deviations for Perplexity Evaluation

We calculate the standard deviations across the 5000 OpenWebText prefixes used for evaluation for Llama-2-7B and Mistral 7B and show them below in Tables 5 and 6. We note that Superposed Decoding has higher standard deviation than Nucleus Sampling and Beam Search when implemented

Table 4: Superposed Decoding is highly competitive with Nucleus Sampling even when only $n$-grams from $n = 2$ to $n = 4$ are interpolated to calculate $p_{\text{ngram}}$.

| | Nucleus | Beam/Greedy | N-Gram | Superposed Decoding | | | |
|---|---|---|---|---|---|---|---|
| Draft # | - | - | - | 1 | 2 | 3 | Best |
| Avg Perplexity | 5.17 | 3.77 | 152.75 | 5.83 | 8.38 | 9.97 | 5.18 |

on Llama-2-7B but that the standard deviations equalize on Mistral 7B. Despite this high variance, human evaluation results in Section 4.4 affirm that Superposed Decoding is consistently competitive in coherency.

Table 5: Standard deviation of Llama-2-7B generation perplexity calculated on OpenWebText test split in Section 4.1.

| | Nucleus | Beam/Greedy | N-Gram | Superposed Decoding | | | |
|---|---|---|---|---|---|---|---|
| Draft # | - | - | - | 1 | 2 | 3 | Best |
| Avg Perplexity | 4.82 | 3.31 | 305.00 | 8.90 | 14.44 | 20.58 | 8.24 |

Table 6: Standard deviation of Mistral 7B generation perplexity calculated on OpenWebText test split in Section 5.1.

| | Nucleus | Superposed Decoding | | | |
|---|---|---|---|---|---|
| Draft # | - | 1 | 2 | 3 | Best |
| Avg Perplexity | 10.19 | 9.92 | 11.53 | 12.53 | 9.38 |

# E   Further Ablations

In this section, we present further ablations studying the impact of number of drafts, prefix length, and generation length on Superposed Decoding perplexity (average best). We find that for scenarios such that the number of drafts $k \geq 3$, Superposed Decoding consistently outperforms nucleus sampling (Figure 9). This is generally expected as the higher the number of drafts, the higher the likelihood that at least one draft is good quality. Similarly, we find that Superposed Decoding is robust to prefix length and generation length. Indeed, the patterns of Superposed Decoding given longer prefix and generation lengths mimic those of Nucleus Sampling in the same situation, with consistently similar perplexities suggesting good performance (Figures 10, 11).

# F   Human Evaluation

## F.1   Additional Results

We run the same experiment as in Section 4.4 two more times at a smaller scale. First, we compare two Nucleus Sampling drafts and three Superposed Decoding drafts. This scenario results in a significant compute advantage for Nucleus Sampling. Still, we find that Superposed Decoding wins the majority of the time, with a win rate of $60.6\%$. However, this is slightly lower than the $64\%$ win rate in the main study.

Next, we compare one Nucleus Sampling draft and the lowest perplexity Superposed Decoding draft of three. This mimics the perplexity evaluation in Section 4.1 but in a human setting. We find that Superposed Decoding is still ranked first the majority of the time, with a $51\%$ win rate.

For cost efficiency, we run both studies on only 100 random prompts, with five responses per prompt.

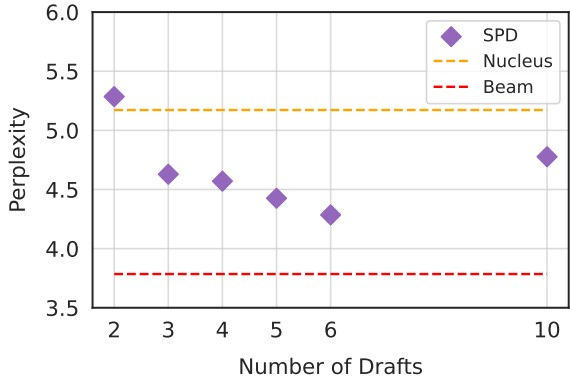

Figure 9: Average perplexity of one draft of Nucleus Sampling and Beam Search compared against average best perplexity of SPD using drafts $k = \{2, 3, 4, 5, 6, 10\}$ on OpenWebText. SPD outperforms Nucleus Sampling for all values of $k$ tested.

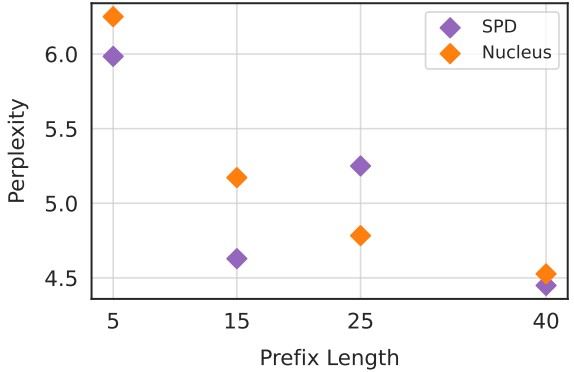

Figure 10: Average perplexity of one draft of Nucleus Sampling compared against average best perplexity of SPD for prefix lengths 5, 15, 25, 40. We find that SPD and Nucleus Sampling consistently have similar perplexities.

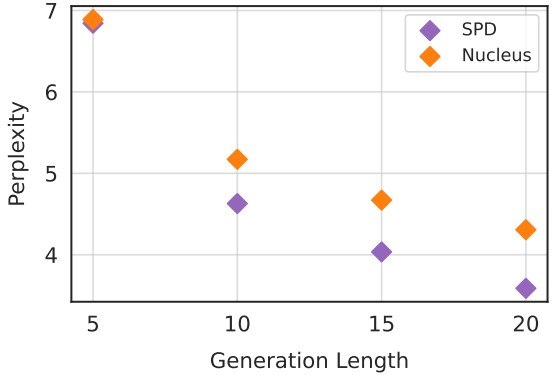

Figure 11: Average perplexity of one draft of Nucleus Sampling compared against average best perplexity of SPD for generation lengths 5, 10, 15, 20. We find that SPD is robust for a wide range of short form generation lengths.

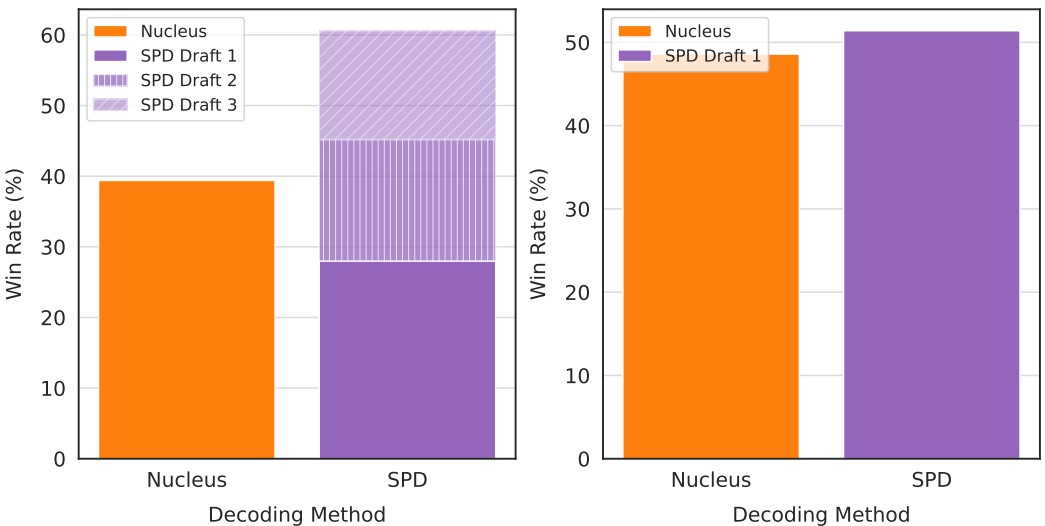

Figure 12: **Left**: Study results comparing two Nucleus Sampling drafts and three Superposed Decoding drafts. **Right**: Study results comparing one Nucleus Sampling draft & the lowest perplexity Superposed Decoding draft out of three generations. Note that uncertainty is higher for both iterations due to smaller sample size.

### F.2 Mechanical Turk Survey

We show a screenshot of our study page in Figure 13. We compensate workers at an approximate rate of $15 per hour.

## G Example Generations

### G.1 Superposed Decoding

In Figure 14, we show 12 additional example generations comparing Superposed Decoding and Nucleus Sampling in a compute-normalized setting. Prefixes are sampled from OpenWebText and truncated to the first 15 tokens, and each generated completion is 10 tokens in length.

### G.2 Nucleus Sampling and Superposed Decoding

In Figure 15, we show an example generation that outputs three drafts using Nucleus Sampling for several timesteps, followed by Superposed Decoding with $k = 3$. This results in nine total drafts at the compute of only three Nucleus Sampling drafts. The final Superposed Decoding drafts maintain the wide coverage offered by Nucleus Sampling and complement it with additional local coverage.

## H Layer-Wise Linearity

In Figure 16, we show the cosine similarity between superposed embeddings $\{\tilde{x}\}$ and the linear combination of the component embeddings of three tokens, after each layer within the first five timesteps using Llama-2-7B. Within each timestep, superposed embeddings pass through an initial layer (Layer 1) that embeds token indices and weights the resulting embeddings using corresponding token weights. The weighted embeddings then pass through 32 transformer blocks (Layers 2-33), then through a final linear projection layer (Layer 34) that produces the output superposed embedding of that timestep.

# I    Resetting

Here, we show examples of the Superposed Decoding Resetting technique described in Section 5.3 for generations of 200 tokens using prefixes from OpenWebText. Superposed Decoding Resetting is significantly less repetitive than non-reset Superposed Decoding. However, Nucleus Sampling, which is also shown, is qualitatively still the best performing decoding method in long form settings.

**Prompt #1**

---

**Superposed Decoding with Resetting**

**Sydney architect Rana Abboud won a NAWIC Scholarship** for 2019. Sydney-based architect and interior designer Rana Abboud has been awarded a $10,000 scholarship to study a Masters of Architecture at the University of New South Wales. The scholarship, awarded by the National Association of Women in Construction (NAWIC) is designed to help women in the construction industry to advance their careers. Rana Abboud, a graduate of the University of Sydney, is the founder of Rana Abboud & Associates. "I am honoured to be awarded this scholarship. I am passionate about design built for people and the environment. I am excited to be able to further my studies and continue to develop my skills and knowledge," she said. The NAWIC Scholarship is awarded to women who are studying or working in the construction industry. "The NAWIC Scholarship is a great opportunity for women to further their care and

---

**Superposed Decoding without Resetting**

**Sydney architect Rana Abboud won a NAWIC Scholarship** in 2017. The National Association of Women in Construction (NAWIC) has announced the winners of its 2017 scholarships. The NAWIC Scholarship Program is designed to encourage women to pursue careers in the construction industry. The 2017 scholarship recipients are: Rana Abboud, a student at the University of New South Wales, who is studying a Bachelor of Architecture. Megan Brennan, a student at the University of New South Wales, who is studying a Bachelor of Construction Management. Katee Cox, a student at the University of New South Wales, who is studying a Bachelor of Construction Management. Kate Dunn, a student at the University of New South Wales, who is studying a Bachelor of Construction Management. Kate Foster, a student at the University of New South Wales, who

---

**Nucleus Sampling**

**Sydney architect Rana Abboud won a NAWIC Scholarship** of $1000 for 'Significant Contribution to Career Growth'. Sydney architect Rana Abboud has won the National Association of Women in Construction (NAWIC) NSW/ACT Scholarship for 2017 for 'Significant Contribution to Career Growth'. She has also just been accepted to undertake a Master of Science in Architecture with Higher Degree by Research at Sydney University. The scholarship is one of eight awards presented by NAWIC to women to further their studies in the construction industry in Australia. "As a working mother, I am very grateful for the support I am receiving to further my professional and academic development," Abboud said. Rana Abboud (middle) with associates (L-R) Tara Abdul-Khalek and Aditi Rana. Abboud is a director of Pal Architecture in Alexandria,

---

**Prompt # 2**

### Superposed Decoding with Resetting

**NEW DELHI: Here's something ayurveda g**urus have been telling for years. A new study has found that a combination of three herbs - including ashwagandha, guduchi and shatavari - can help in the treatment of diabetes. "The study was conducted to evaluate the efficacy and safety of a poly-herbal formulation containing three herbs - Ashwagandha (Withania somnifera), Guduchi (Tinospora cordifolia), and Shatavari (Asparagus racemosus) in inhibiting the progression of diabetes in alloxan-induced diabetic rats" said the study published in the journal 'Journal of Ayurveda and Integrative Medicine'. The study was conducted by research team from the Department of Pharmacology, Dr D. Patil College of Pharmacy, Pune, Maharashtra, and the Department of Pharm

### Superposed Decoding without Resetting

**NEW DELHI: Here's something ayurveda g**urus have been saying for years: Eat your way to health. A new study has found that a diet rich in fruits and vegetables can help reduce the risk of developing type 2 diabetes. The study, published in the journal BMJ Open, found that a diet rich in fruits and vegetables can help reduce the risk of developing type 2 diabetes. The study, which was conducted by researchers from the University of Oxford, found that a diet rich in fruits and vegetables can help prevent the risk of developing type 2 diabetes. The study, which was conducted by researchers from the University of Oxford, found that a diet rich in fruits and vegetables can help prevent the risk of developing type 2 diabetes. The study, which was conducted by researchers from the University of Oxford, found that a diet rich in fruits and vegetables can help prevent the risk

### Nucleus Sampling

**NEW DELHI: Here's something ayurveda g**urus have been telling us about for years - eating seasonal fruits and vegetables helps stay healthy. Here are 7 foods which can keep you healthy during the summer. Pomegranate is a summer fruit and if you wish to have optimum health, it is necessary to eat this fruit. The fruit can be eaten in raw form, or cut into pieces and mixed with honey and taken twice a day. The skin of the fruit is itself a health tonic and can be chewed. It contains natural bleaching properties. The fruit can also be squeezed and then the pulp mixed with milk, cumin seeds, one teaspoonful of honey and a pinch of ginger and drank thrice daily. Beets are a popular root vegetable. Beetroot juice may help increase exercise endurance, potentially because it increases blood flow, said WebMD. The best fruits

**Prompt # 3**

### Superposed Decoding with Resetting

**As part of a broad initiative to combat sexual harassment and** assault, the University of California, Berkeley has launched a new website to help students, staff, and faculty report and respond to sexual harassment and assault. The new site, which went live on Monday, is the one-stop destination for all things related to sexual harassment and assault at UC Berkeley. It is the result of a collaboration between the Office of the Vice Chancellor of Student Affairs and the Office of the Vice Provost for Academic and Student Affairs. "This is a critical time for our campus and to come together to address sexual harassment and assault in our community," said Vice Chancellor for Student Affairs, Steph Sutton-Wheeler, in a statement. "We are committed to providing a safe and inclusive environment for all members of our community and this new website is a critical step in that effort." The site provides information on how to report sexual harassment and assault, as well as the for

### Superposed Decoding without Resetting

**As part of a broad initiative to combat sexual harassment and** assault, the University of California, Berkeley, has launched a new website that provides information about the university's policies and procedures for reporting and responding to sexual harassment and sexual violence. The website, which is available at http://sexualharassment.berkeley.edu, is designed to help students, faculty and staff understand their rights and responsibilities under the university's policies and procedures. The website includes information about the university's policies and procedures for reporting and responding to sexual harassment and sexual violence, as well as information about the university's policies and procedures for investigating and responding to sexual harassment and sexual violence. The website also includes information about the university's policies and procedures for investigating and responding to sexual harassment and sexual violence, including information about the university's policies and procedures for investigating and responding to sexual harassment and sexual violence in the workplace.

### Nucleus Sampling

**As part of a broad initiative to combat sexual harassment and** misconduct on campus, a task force consisting of faculty, staff, students, and other members of the campus community will present a draft report with recommendations at the Board of Trustees meeting on Friday, October 27, 2017. Under the leadership of Executive Director for Human Resources Jesse Bozeman, the task force was formed to address concerns raised at the end of the Spring 2017 semester, which included a Student Government Association resolution and a student-driven, faculty-approved course called "Sexual Misconduct: Consent and Culture," taught by Professor Nancy Cantalupo and co-sponsored by Campus Life, Student Affairs, and Academic Affairs. The task force is comprised of the following members: Diana Ballin, Senior Lecturer in the Department of Anthropology, Chair; Patricia Hopkins, Dean of Students; Kelly McSweeney

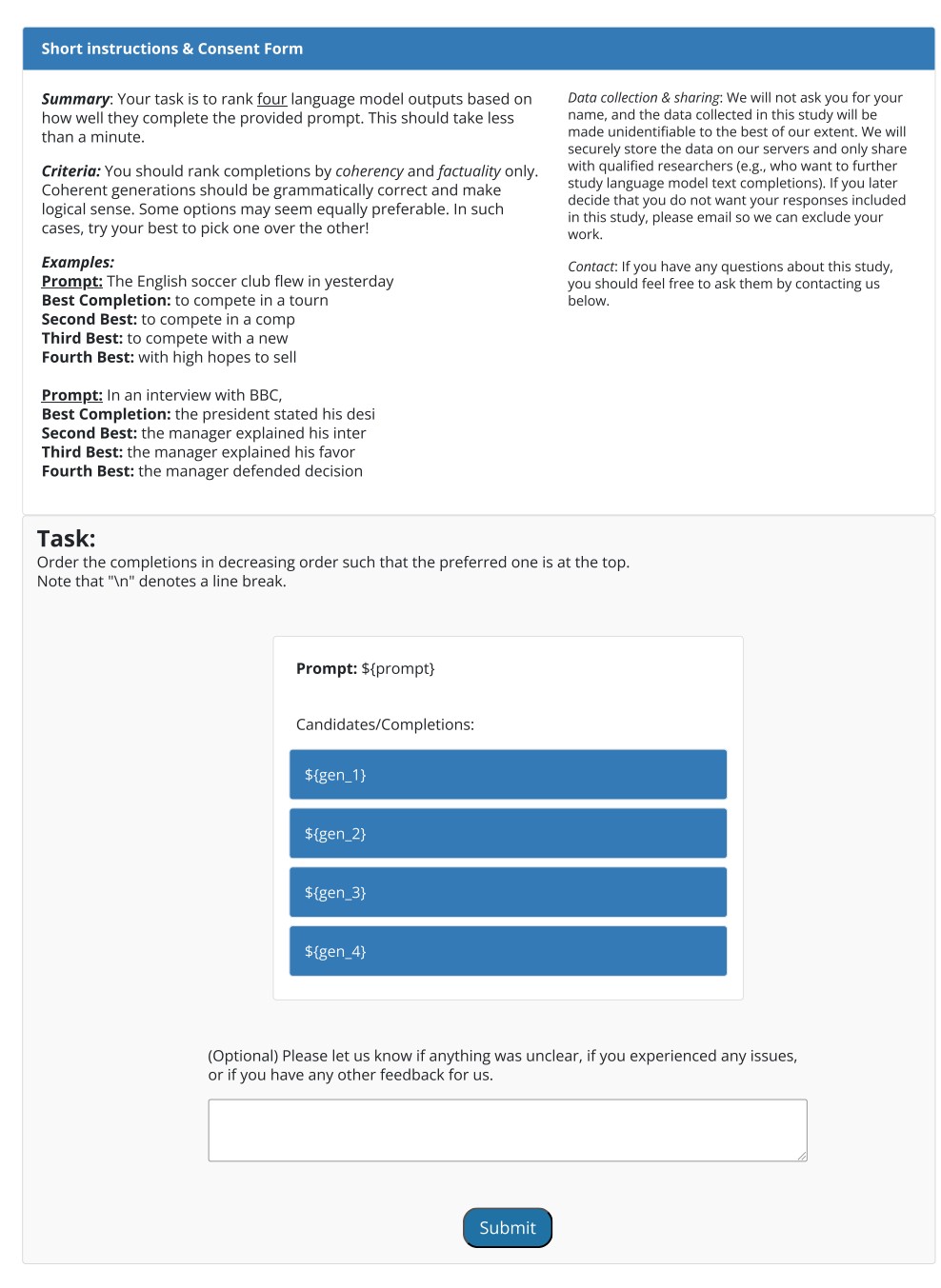

**Short instructions & Consent Form**

**Summary**: Your task is to rank four language model outputs based on how well they complete the provided prompt. This should take less than a minute.

**Criteria:** You should rank completions by *coherency* and *factuality* only. Coherent generations should be grammatically correct and make logical sense. Some options may seem equally preferable. In such cases, try your best to pick one over the other!

**Examples:**
Prompt: The English soccer club flew in yesterday
**Best Completion:** to compete in a tourn
**Second Best:** to compete in a comp
**Third Best:** to compete with a new
**Fourth Best:** with high hopes to sell

Prompt: In an interview with BBC,
**Best Completion:** the president stated his desi
**Second Best:** the manager explained his inter
**Third Best:** the manager explained his favor
**Fourth Best:** the manager defended decision

*Data collection & sharing*: We will not ask you for your name, and the data collected in this study will be made unidentifiable to the best of our extent. We will securely store the data on our servers and only share with qualified researchers (e.g., who want to further study language model text completions). If you later decide that you do not want your responses included in this study, please email so we can exclude your work.

*Contact*: If you have any questions about this study, you should feel free to ask them by contacting us below.

**Task:**
Order the completions in decreasing order such that the preferred one is at the top.
Note that "\n" denotes a line break.

**Prompt:** ${prompt}

Candidates/Completions:

${gen_1}

${gen_2}

${gen_3}

${gen_4}

(Optional) Please let us know if anything was unclear, if you experienced any issues, or if you have any other feedback for us.

Submit

Figure 13: A screenshot of our Mechanical Turk template, which presents the respondent with a task summary, objective criteria for ranking generations, and two examples.

| | | When I worked as a scout for the Carolina Panthers in the | As a kid, I played a game called The Sims, a | Over the next few years, NASA plans to get back to launching |
|---|---|---|---|---|
| | OpenWebText | | | |
| | Nucleus | 1990s, I would often | simulation of life in a suburban neighborhood. I | rockets from Cape Canaveral, Florida |
| | Superposed | 1990s, I was **always**
1990s, I was **a**
1990s, I was **responsible** | game where you **can** create your character and live **out**
game where you **could** create a character and live **out**
game where you **can** create your character and live **their** | astronauts from U.S. soil**.**
astronauts from U.S. soil**,**
astronauts from U.S. soil **for** |

| | | "Do you want an extended warranty with that?" | Synopsis A library for running tasks(jobs) on | Just this Tuesday I wrote calling on MUD to present a |
|---|---|---|---|---|
| | OpenWebText | | | |
| | Nucleus | "No, thanks. I'm sure it | a cluster of nodes. It provides a simple API | plan to the community to resolve the situation with the |
| | Superposed | "No, I don't **want an** extended
"No, I don't **think an** extended
"No, I don't **want any** extended | a cluster of **machines**. ## Installation
a cluster of **computers**. ## Installation
a cluster of **nodes**. ## Installation | plan to the **public** for the future of the **M**
plan to the **public** for the future of the **site**
plan to the **community** for the future of the **M** |

| | | As an undergraduate, I took a course titled "The Log | "Guess what? I have flaws. What are they? | In this gluten-free riff on banana cream pie |
|---|---|---|---|---|
| | OpenWebText | | | |
| | Nucleus | ic of Scientific Discovery" with Karl Po | I'm a little too bossy, | , the banana flavor comes from a hom |
| | Superposed | ic of Scientific Discovery" **with** the late
ic of Scientific Discovery" **in** the late
ic of Scientific Discovery" **from** the late | I'm not **sure**. I'm **working**
I'm not **sure**. I'm **not**
I'm not **perfect**. I'm **working** | , the bananas are baked into **a cust**
, the bananas are baked into **a cr**
, the bananas are baked into **the cust** |

| | | As most managers are probably starting to figure out, Midfielders | Martin Shkreli, the bad boy of the U.S. | Various ancient maps discovered throughout time reveal a slightly different reality than |
|---|---|---|---|---|
| | OpenWebText | | | |
| | Nucleus | are the most important part of a team. This | pharmaceutical industry, was convicted | what we are used to seeing. The maps |
| | Superposed | are the most important **part** of **the** team. They
are the most important **players** of **the** team. They
are the most important **part** of **a** team. They | pharmaceutical industry, has been **sent**
pharmaceutical industry, has been **found**
pharmaceutical industry, has been **arrested** | what we are used to. The **most famous**
what we are used to. The **world famous**
what we are used to. The **most interesting** |

Figure 14: Additional qualitative text generations in a compute-normalized setting for Superposed Decoding and Nucleus Sampling with prefixes sampled from OpenWebText

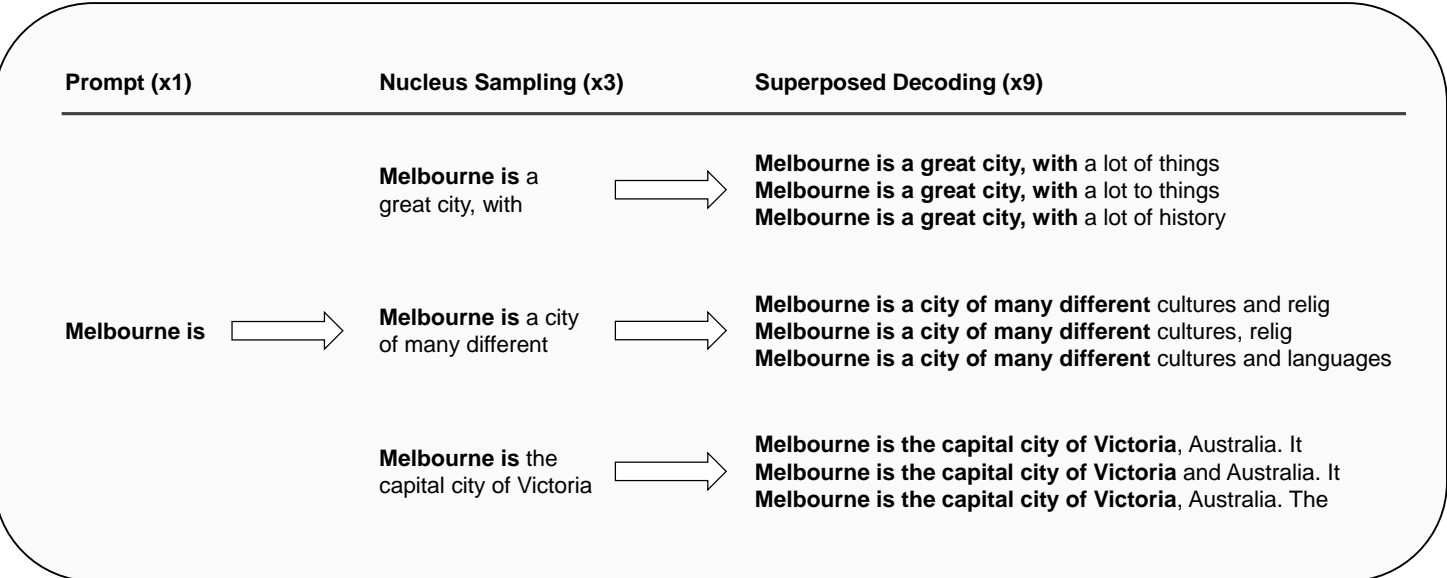

Figure 15: Example output generated by combining Nucleus Sampling and Superposed Decoding. Superposed Decoding generates three drafts per Nucleus Sampling sample, resulting in nine total drafts.

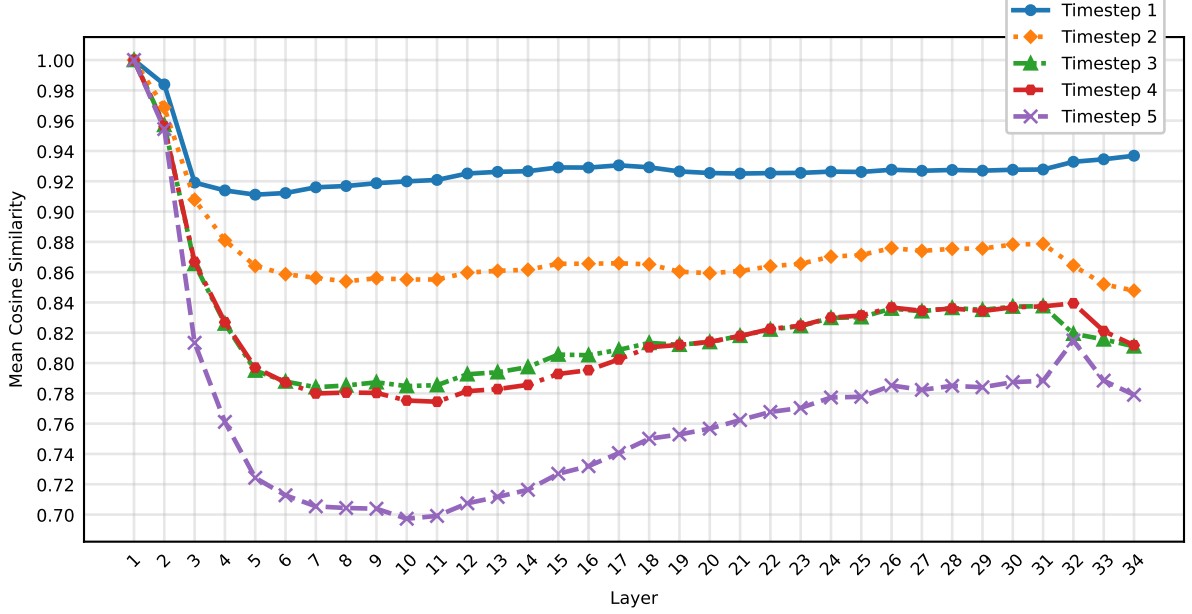

Figure 16: Layer-wise linearity analysis of the first five timesteps on Llama-2-7B with three tokens. The relationship between superposed embeddings and the component token embeddings is initially entirely linear; linearity then degenerates over the first few layers, but gradually recovers through the subsequent transformer block layers.

