# OpenReview forum: "Superposed Decoding: Multiple Generations from a Single Autoregressive Inference Pass"
_NeurIPS.cc/2024/Conference — NeurIPS 2024 poster_

### Official Review · Reviewer_Lot4 · 2024-07-10

**Soundness:** 4
**Presentation:** 3
**Contribution:** 3
**Rating:** 7
**Confidence:** 3

**Summary:**

This paper introduces Superposed Decoding, a novel method to generate multiple drafts (k drafts) in a single inference pass. The process involves two iterative steps: (1) running a large language model (LLM) inference on fused tokens and utilizing top-k sampling to produce k candidate tokens, and (2) combining n-gram probability scores with LLM probability scores to extend each draft with one candidate token, iteratively. This method demonstrates significant improvements in generation perplexity, downstream task performance, and is favored by human annotators, delivering results 2.44 times faster. The authors also suggest a resetting mechanism to mitigate repetitions in long drafts.

**Strengths:**

- The method is intelligent and innovative, addressing a task that, while not highly popular, has numerous practical applications and significant real-world relevance.
- The experiments are well-designed, covering text generation, question answering, and human evaluation, showcasing the method's effectiveness in both quality and efficiency.

**Weaknesses:**

- The current application is restricted to text drafting and does not extend to code generation, which could also benefit from this method and is frequently used in practice.

**Questions:**

- In Section 5.2, the authors indicate that shorter generations lead to higher diversity as shown by Self-BLEU. How does this diversity change in other decoding methods? Do these methods exhibit similar diversity patterns as Superposed Decoding?

Overall, I lean towards accepting this work. The weaknesses and questions are primarily around clarifications rather than flaws in the method. If the authors address these questions and provide the necessary clarifications, I am inclined to raise my score.

**Limitations:**

The authors acknowledge and provide reasonable limitations:

- The quality of Superposed Decoding largely depends on the n-gram models used, which are vital for maintaining coherence.
- While the drafts produced are syntactically diverse, they may lack semantic diversity.

---

> ### Author Rebuttal · Authors · 2024-08-05
>
> We thank the reviewer for the positive review. We are glad that the reviewer found the approach innovative and experiments well designed. Following are the clarifications requested in the review:
>
> **1. Code generation:** We agree with the reviewer and mention in the manuscript that code generation is yet another important step use case. However, in this paper we focus on language generation to showcase the generality of the method and hope to extend it to code generation in the future.
>
> **2. Diversity of other decoding schemes for short generations:** We run additional experiments generating three drafts using Beam Search and Nucleus Sampling and calculate the diversity of drafts using Self-BLEU. We find that while beam search, like Superposed Decoding, becomes more diverse at shorter generation lengths, the opposite is true with Nucleus Sampling. We attribute this to the fact that Beam Search, like Superposed Decoding, produces syntactically different but semantically alike drafts. This means that as generation length increases, more tokens are generally in common between drafts. On the other hand, Nucleus Sampling’s probabilistic nature results in semantically different drafts. Consequently, longer generation lengths lead to a higher proportion of token-level differences.
>
> _Diversity as measured by Self-BLEU_
> | Generation Length        | 5    | 10   | 15   | 20   |
> |---------|------|------|------|------|
> | **Superposed Decoding** | 0.51 | 0.81 | 0.88 | 0.91 |
> | **Beam**   | 0.35 | 0.64 | 0.75 | 0.81 |
> | **Nucleus** | 0.33 | 0.25 | 0.21 | 0.19 |
>
> We hope that this rebuttal solidifies your positive outlook on the paper and we are happy to discuss if you need any further clarifications to increase the score and facilitate acceptance of the paper.

---

> > ### Comment · Reviewer_Lot4 · 2024-08-13
> >
> > Thanks for the authors' response. After carefully reading the response and the reviews from other reviewers, I would like to keep the score unchanged and recommend acceptance.

---

> ### Author Response · Authors · 2024-08-13
>
> Thank you for your response and the support for the acceptance of the paper.

---

### Official Review · Reviewer_ZGM9 · 2024-07-10

**Soundness:** 2
**Presentation:** 3
**Contribution:** 2
**Rating:** 4
**Confidence:** 4

**Summary:**

The paper "Superposed Decoding: Multiple Generations from a Single Autoregressive Inference Pass" introduces a method to generate
k similar responses in parallel using a single forward pass in autoregressive models.

**Strengths:**

The paper validates its method on two open-source LLMs, providing empirical evidence of its applicability.

**Weaknesses:**

1. Limited Problem Scope and Optimization Potential:

The problem addressed might not have significant optimization potential. Using sampling and batch construction during inference can prevent an increase in latency. Autoregressive decoding is often memory-bounded, so the batch inference will not be a big burden for latency. It is unclear if the authors have considered batch construction for the vanilla method when measuring latency.

2. Restricted Scenario and Lack of Strong Justification:

The scenario studied is highly restricted, and the authors do not provide compelling evidence that the proposed method yields superior results for n-decoding. Specifically, the top-1 result obtained using the proposed method performs worse than vanilla decoding.

3. Complexity and Additional Dependencies:

The proposed method appears inelegant, relying on an additional n-gram language model to calibrate the outputs of the LLM. This dependency detracts from the method's appeal and does not demonstrate a clear advantage over batch inference.

**Questions:**

see weakness

---

> ### Author Rebuttal · Authors · 2024-08-05
>
> In the following we provide clarifications that were requested:
>
> **1. Limited scope and restricted scenarios:** We respectfully disagree with the reviewer about the limited scope and scenarios of Superposed Decoding. Short text generation and drafting are common real-world problems and a centerpiece to products such as GitHub Copilot and Tabnine. Features like Gmail SmartCompose, Word auto-suggestions, and iPhone auto-completions also rely on providing multiple suggestions.
>
> **2. Optimization potential:** We agree with the reviewer that batch construction helps with latency and that autoregressive decoding is often memory bound. However, solely using batch construction for multiple drafts reduces effective batch size by a factor of $k$ (i.e. number of prefixes that can be handled at once). On the other hand, Superposed Decoding is agnostic to these challenges and _complementary to batch construction_. Superposed Decoding helps generate $k$ drafts for every decoding run, _effectively multiplying the batch size of any decoding scheme_ like nucleus sampling. This avoids the limitations imposed by vanilla batching, benefiting scenarios where batch size is a limitation, such as server-based applications like Github Copilot or Tabnine.
> An interesting way to think about Superposed Decoding is as a way to execute local searches while each nucleus sampling draft would be a global search. One can always combine both ideas and benefit from increasing the effective number of generated drafts. We are happy to discuss this further.
>
> **3. Lack of strong justification:** It is true that the top-1 result from Superposed Decoding is slightly outperformed by vanilla decoding in some metrics. However, these differences are small: 0.04% on TriviaQA and 0.33% on Natural Questions. In addition, Superposed Decoding is primarily meant for *multiple draft scenarios*, where it can significantly outperform the other decoding methods without *increasing compute* (Figure 5, Figure 7). In the case where the goal is just a single generation, then vanilla decoding (or any other sampling method) can be used at no detriment by simply toggling off n-gram filtering.
>
> **4. Complexity and dependence on n-gram models:** We respectfully disagree that n-gram models limit the convenience of Superposed Decoding. Data stores and non-parametric modeling are widespread in language models (ACL 2023 Tutorial on Retrieval-based Language Models and Applications). For example, “Nonparametric Masked Language Modeling” establishes the use of an external corpus of two billion tokens for masked language generation (Min et al., ACL 2023). Similarly, large text corpuses are essential for document retrieval and RAG applications (Karpukhin et al., EMNLP 2020; Shi et al., NAACL 2024). N-gram models have also proven to be powerful in machine translation settings (Branst et al., ACL 2007). Finally, with the advances like Infinigram (Liu et al., COLM 2024), n-gram datastores are easy to access and scale up as part of language modeling.
>
> We hope that this rebuttal addresses your concerns and we are happy to discuss further if you need any further clarifications to increase the score and facilitate acceptance of the paper.

---

> > ### Comment · Reviewer_ZGM9 · 2024-08-08
> >
> > My major concern is still the n-gram LM, your rebuttal is about all the nonparametric LMs. However, we are discussing n-gram LM. And the evidence you gave like Branst et al., ACL 2007 is a very old setting. I am an expert in machine translation, and I think this evidence can not prove anything in the context of LLM.
> >
> > And I'm very concerned that your method can only find some paraphrased sentences rather than covering different modes of the data like top-p sampling. The self-bleu also validated my concern.
> >
> > Can you add more experiments like [1].
> >
> > [1] Large Language Monkeys: Scaling Inference Compute with Repeated Sampling, arxiv

---

> > > ### Author Response · Authors · 2024-08-09
> > >
> > > We thank the reviewer for their prompt reply. Here are our thoughts.
> > >
> > > **1. Nonparametric LMs:** We respectfully disagree that previous non-parametric LM research is unrelated to our method. In the context of language models, non-parametric language models are defined as language models whose data distribution is “a function of the available data,” with complexity growing with data (Siegel, 1957; Hollander et al., 2013; Min et al., 2023). We believe that this definition well-encapsulates a language model interpolated with n-gram models. Indeed, we see similar use cases in kNN-LM (Khandelwal et al., 2020) and NPM (Min et al., 2023), which use external datastores that behave like n-gram models - *the sole difference is that they are queried with a representation vector instead of a token sequence.*
> > >
> > > Superposed Decoding also can rely on representation vectors for retrieval from a datastore rather than directly doing a dictionary lookup during n-gram filtering. The former is “semantic” while the latter is “syntactic”.
> > >
> > > **2. Modes of data:** Yes, the reviewer is right about Superposed Decoding potentially being unable to cover significantly different modes of the data like top-p sampling. However, this is not a concern nor a bug. The goal of Superposed Decoding is to provide multiple drafts at constant compute to help increase coverage. This coverage aids in improving user experience and factuality as shown in the experiments. As we mention, Superposed Decoding is intended for local exploration. More often than not, the greedy top-1 generation solves the problem at hand; further enabling local search and multiple drafts around it supports quite a few use cases.
> > >
> > > Having said that, Superposed Decoding can be combined with any sampling method, be top-p or something else, and it can help generate a local set of drafts for each sampling trajectory at no additional compute cost. For instance, if top-p sampling is used to generate $n$ drafts, Superposed Decoding with $k$ local drafts can be spliced in at any timestep to strategically produce $nk$ drafts, where each top-p trajectory is bolstered by $k$ local explorations at no extra cost. We include a sample generation below, using Superposed Decoding to produce three local drafts for each top-p sample; this expands the available options without extra compute or reducing mode coverage.
> > >
> > > *SPD denotes Superposed Decoding.*
> > > ```
> > > Example of Top-p Sampling w/Superposed Decoding:
> > > │  Prefix: “Melbourne is”
> > > │
> > > └───Top-p: Melbourne is a great city, with
> > > │   │   SPD: Melbourne is a great city, with a lot of things
> > > │   │   SPD: Melbourne is a great city, with a lot to things
> > > │   │   SPD: Melbourne is a great city, with a lot of history
> > > │
> > > └───Top-p: Melbourne is a city of many different
> > > │   │   SPD: Melbourne is a city of many different cultures and relig
> > > │   │   SPD: Melbourne is a city of many different cultures, relig
> > > │   │   SPD: Melbourne is a city of many different cultures and languages
> > > │
> > > └───Top-p: Melbourne is the capital city of Victoria
> > > │    │   SPD: Melbourne is the capital city of Victoria, Australia. It
> > > │    │   SPD: Melbourne is the capital city of Victoria and Australia. It
> > > │    │   SPD: Melbourne is the capital city of Victoria, Australia. The
> > > ```
> > >
> > > Do let us know if we are missing something here, and we are happy to discuss further. Also, let us know if the concern about optimization potential was resolved through the rebuttal.

---

> > > > ### Comment · Reviewer_ZGM9 · 2024-08-09
> > > >
> > > > Can you add more experiments like [1].
> > > >
> > > > [1] Large Language Monkeys: Scaling Inference Compute with Repeated Sampling, arxiv
> > > >
> > > > If cannot, I still do not think the proposed method has a clear practical usage.

---

> ### Author Response · Authors · 2024-08-12
>
> We ran experiments from [1] and found that **Superposed Decoding significantly improves accuracy with no additional compute**, demonstrating its practicality. On TriviaQA and Natural Questions, **combining Nucleus Sampling and Superposed Decoding results in better performance** than vanilla Nucleus Sampling across the board.
>
> TriviaQA (the three rows in each column use the same compute):
>
> | Compute ($k$) | 1         | 10        | 20        | 30        | 40        | 50        | 60        | 70        | 80        | 90        | 100       |
> |---------------|-----------|-----------|-----------|-----------|-----------|-----------|-----------|-----------|-----------|-----------|-----------|
> | $NS$          |     51.04 |     68.75 |     70.31 |     71.87 |     72.92 |     74.48 |     74.74 |     75.26 |     75.78 |     76.30 |     76.56 |
> | $NS_{SPD2}$   |     51.30 |     68.75 |     70.57 |     72.66 |     74.74 |     75.78 |     76.30 |     76.82 |     78.39 |     79.17 |     79.43 |
> | $NS_{SPD3}$  | **51.82** | **70.57** | **74.22** | **75.52** | **77.34** | **77.87** | **78.39** | **78.65** | **79.17** | **79.43** | **79.95** |
>
> Natural Questions (the three rows in each column use the same compute):
>
> | Compute ($k$) | 1         | 10        | 20        | 30        | 40        | 50        | 60        | 70        | 80        | 90        | 100       |
> |---------------|-----------|-----------|-----------|-----------|-----------|-----------|-----------|-----------|-----------|-----------|-----------|
> | $NS$          |     14.32 | **32.55** | **36.98** |     38.54 |     40.36 |     41.15 |     41.67 |     41.93 |     42.19 |     42.71 |     42.97 |
> | $NS_{SPD2}$   |     15.36 |     31.25 |     34.90 |     38.02 |     39.84 |     41.41 |     41.67 |     42.45 |     43.75 |     43.75 |     43.75 |
> | $NS_{SPD3}$   | **15.63** |     31.25 | **36.98** | **39.06** | **41.15** | **42.71** | **43.75** | **43.75** | **44.27** | **44.79** | **45.57** |
>
> Experimental Setup ($NS$: Nucleus Sampling; $SPD$: Superposed Decoding):
> - We evaluate three decoding strategies on Llama-2-7B:
>    - Vanilla Nucleus Sampling for $n$ timesteps ($NS$).
>    - Nucleus Sampling for $c$ timesteps then two SPD drafts for $n-c$ timesteps ($NS_{SPD2}$).
>    - Nucleus Sampling for $c$ timesteps then three SPD drafts for $n-c$ timesteps ($NS_{SPD3}$).
> - We compare accuracy on 1-100 inputs in a constant compute setting. This means the performance of the first $k$ $NS$ samples is compared against the first $2k$ $NS_{SPD2}$ samples and $3k$ $NS_{SPD3}$ samples, where $k \leq 100$.
> - In the tables above, each column denotes results given the same compute. We show accuracy at intervals of 10 for $k$. We average results over three runs for each strategy.
>
> We hope that these experiments resolve your concerns about Superposed Decoding’s practicality. We would also like to know if our earlier responses addressed your concerns about our method’s optimization potential, justification, and use of n-gram models.
>
> [1] Large Language Monkeys: Scaling Inference Compute with Repeated Sampling, arxiv

---

> > ### Comment · Reviewer_ZGM9 · 2024-08-13
> >
> > Thanks for your reply. What is the evaluation metric for this table?
> >
> > Do you have an external reward model to perform best-of-n or do you use self-consistency?

---

> > ### Comment · Reviewer_ZGM9 · 2024-08-13
> >
> > And what is $c$? I do not see $c$ in the table, is it $SPD_{c}$?

---

> > > ### Comment · Reviewer_ZGM9 · 2024-08-13
> > >
> > > Please respond to my question as soon as possible to clear up any misunderstanding I have about your rebuttal.

---

> ### Author Response · Authors · 2024-08-13
>
> Following are the requested clarifications:
>
> **1. Evaluation Metric:** We use precision as our metric. In addition, we follow the main body of [1] (Sections 2 and 3) and assume an ideal scenario where the best sample can always be identified. While [1] does suggest the usage of other verification methods, the paper’s experiments are primarily conducted assuming an ideal scenario.
>
> **2. Notation:** $c$ is the number of timesteps that are generated with Nucleus Sampling before Superposed Decoding is applied to create drafts. $n$ is the total number of timesteps. For example, in TriviaQA, $c = 4$ and $n = 10$. For Natural Questions, $c = 9$ and $n = 15$. These values are held constant.
> In the table, $NS_{SPD2}$ and $NS_{SPD3}$ denote whether two or three drafts are generated after $c$ timesteps.
>
> Please let us know if anything is still unclear and thanks for extremely prompt responses.
>
> [1] Large Language Monkeys: Scaling Inference Compute with Repeated Sampling, arxiv

---

> > ### Comment · Reviewer_ZGM9 · 2024-08-13
> >
> > Why do you use this setting instead of just use only SPD?

---

> > > ### Author Response · Authors · 2024-08-13
> > >
> > > As we mentioned earlier in the rebuttal (and see Modes of Data in [here](https://openreview.net/forum?id=KSOkkHm9I7&noteId=T6Le0oX7hh)), SPD enables local search that helps improve the coverage of generations at no additional cost. For example, one nucleus sampled generation can be expanded by $k=3$ times using Superposed Decoding without increasing the compute. We do not claim that three SPD drafts are as good as three Nucleus Sampling drafts, but only that they are better than only one nucleus sampled draft, as shown in our paper’s experiments. This is a fair comparison because three nucleus sampled drafts will cost three times the compute as three Superposed drafts.
> > >
> > > Extending this further to the experiment above, we want to show that Superposed Decoding helps cover many modes of data. Using Nucleus Sampling as the scaffold (which can span the global space) and then generating many more drafts for the same compute cost using Superposed Decoding shows the practicality and complementary nature of SPD. Generating 3x or 100x drafts with SPD will not make a difference after a point (similar to Nucleus Sampling after 20-50 drafts despite linearly increasing compute) as accuracy saturates around a local minima. In this case, SPD helps Nucleus Sampling do a better local search (than just itself) to hit the right mode of the data to accurately answer a given question. This is shown by an asymptotic increase of accuracy/precision by roughly $3$% on both TriviaQA and Natural Questions.
> > >
> > > Do let us know if you have further questions. In short, we present Superposed Decoding as a novel alternative that generates multiple drafts for the same compute cost as a single draft while increasing the coverage for both factuality and user preferences.

---

> > > > ### Comment · Reviewer_ZGM9 · 2024-08-13
> > > >
> > > > The rebuttal partially solved my concern, but I still feel not sure about the importance of coverage of local modes. I raised the score and still welcome the authors to give further justification.

---

> > > > > ### Author Response · Authors · 2024-08-13
> > > > >
> > > > > We thank the reviewer for raising their score. We would like to reiterate several points that highlight the benefits of Superposed Decoding and local mode coverage:
> > > > >
> > > > > - In human evaluation experiments (Figure 7 in the paper), Superposed Decoding’s ability to provide multiple drafts results in generations preferred over Nucleus Sampling (**63% to 37%**). This can considerably benefit user experience in applications such as Github Copilot, Tabnine, Word Auto-Complete, and more.
> > > > >
> > > > > - Alone, Superposed Decoding’s local coverage (three drafts) increases accuracy by **8%** in TriviaQA and **5%** in Natural Questions compared to one Nucleus Sampling generation at the same compute (Figure 5), highlighting significantly better factuality.
> > > > >
> > > > > - Superposed Decoding added to Nucleus Sampling uses local coverage to expand Nucleus Sampling’s global modes, generating more factual outputs at scale than Nucleus Sampling alone without increasing compute cost for 1-100 inputs. This results in consistent accuracy gains up to **3%** as shown [here](https://openreview.net/forum?id=KSOkkHm9I7&noteId=DGv4vjS6rr).
> > > > >
> > > > > We believe Superposed Decoding is an interesting phenomenon that adds value to the community, benefits users, and provides significant accuracy boosts. We are happy to answer any further questions.

---

### Official Review · Reviewer_E6Fn · 2024-07-12

**Soundness:** 1
**Presentation:** 3
**Contribution:** 2
**Rating:** 4
**Confidence:** 4

**Summary:**

This paper proposed a novel decoding algorithm to generate k coherent drafts in one autoregressive inference pass. An additional n-grams model is used to keep the k-drafts coherent. Experimental results show that this method can generate three relatively coherent drafts while achieving a speed-up ratio of more than 2.44.

**Strengths:**

1. This paper is well written.
2. Comprehensive experiments are conducted to evaluate the performance of the proposed method.

**Weaknesses:**

1. This paper proposed Superposed Decoding, which aims to generate k different sequences in a single inference of the LM.
It uses a superposition to approximately represent the last tokens of k drafts in each decoding step. However, it seems that this goal can be easily achieved by adopting tree attention masks [1] without the proposed superposition. For example, given a prefix [x1,x2,...,xn], we can greedy sample k tokens in the output distribution to initialize k different drafts. Next, we concatenate these k tokens with the prefix. Here, the current sequence has n+k tokens. The position id is [1,2,....,n]+[n+1]*k. We produce the corresponding tree attention mask to keep the k drafts independent. If we just want to complete these k drafts, we can input these k tokens in parallel and perform greedy sampling on each of them in each step. This generation also only costs one decoding process. (Let L be the max length of the k drafts; the whole generation process takes L decoding steps.) However, the k drafts are exactly the same as those generated separately.
In contrast, the proposed method loses precision. What is the strength of Superposed Decoding compared to the approach above?

Reference：
[1] Li Y, Wei F, Zhang C, et al. Eagle: Speculative sampling requires rethinking feature uncertainty[J]. arXiv preprint arXiv:2401.15077, 2024.

2. The coherence of the generated drafts depends on the quality of the n-grams model. Meanwhile, the introduction of additional n-gram models limits the convenience of this approach.

**Questions:**

1. See weakness 1.
2. How often does this method generate incoherent or erroneous drafts?

**Limitations:**

The n-grams model is constructed by open-source texts, which containing toxic data that may affect the safety of the model. This factor should be taken into consideration.

---

> ### Author Rebuttal · Authors · 2024-08-05
>
> We are glad that the reviewer found the paper to be well written and are happy to hear that the reviewer appreciated the experiments. Below are the clarifications requested in the review:
>
> **1. Comparison to tree attention masks:** While this paper is interesting and relevant, we do not believe it is a replacement for Superposed Decoding. It is true that tree masking will reduce KV cache size compared to vanilla drafting techniques because the prefix is stored only once. However, there are still _additional storage costs_ from storing every generated token in memory (Cai et al., ICML 2024), which Superposed Decoding does not need to do. These costs are present regardless of batching and can reduce effective batch size (i.e. number of prefixes that can be handled at once). This is further accentuated when the initial prefix length is small compared to the generation length, leading the generated tokens to dominate the overall storage requirement. Furthermore, every additional token that must be stored and processed means less efficiency from a FLOPs perspective. Superposed Decoding does not have any of these limitations because it combines multiple drafts into a single superposed LM input. In addition, Superposed Decoding is _completely complementary_ to Tree Attention because some drafted tokens can be superposed to save memory, only requiring slight adjustment to the tree attention mask. We are happy to discuss further on this.
>
> **2. Dependence on n-gram models:** We respectfully disagree that n-gram models limit the convenience of Superposed Decoding. Data stores and non-parametric modeling are widespread in language models (ACL 2023 Tutorial on Retrieval-based Language Models and Applications). For example, “Nonparametric Masked Language Modeling” establishes the use of an external corpus of two billion tokens for masked language generation (Min et al., ACL 2023). Similarly, large text corpuses are essential for document retrieval and RAG applications (Karpukhin et al., EMNLP 2020; Shi et al., NAACL 2024). N-gram models have also proven to be powerful in machine translation settings (Branst et al., ACL 2007). Finally, with the advances like Infinigram (Liu et al., COLM 2024), n-gram datastores are easy to access and scale up as part of language modeling.
>
> **3. Incoherent and erroneous drafts:** Thank you for this question. While incoherency is an important metric, there are currently no good ways to measure at scale without using proxies like perplexity, which we show in the paper. In practice, we rarely see any incoherent drafts, as highlighted by strong human evaluation performance. Erroneous drafts are a different problem because they are also tied to factuality – which is typically helped by non-parametric modeling and data stores similar to what we do with n-gram filtering.
>
> **4. Toxicity:** The risk of toxicity can be significantly reduced by building n-gram models on clean data stores created by domain experts. Furthermore, n-gram models are frequently used in personalized scenarios. In such cases, the relevant data stores are often pre-filtered to be appropriate for a user or specific task.
>
> We hope that this rebuttal clarifies your concerns, and we are happy to discuss any further clarifications to increase the score and facilitate acceptance of the paper.

---

> > ### Comment · Reviewer_E6Fn · 2024-08-13
> >
> > Thanks for the explanations. I will raise my score to 5.

---

> ### Author Response · Authors · 2024-08-12
>
> We wanted to check in to see if our rebuttal resolved your concerns. We are happy to discuss further.

---

> ### Author Response · Authors · 2024-08-13
>
> We thank the reviewer for agreeing to raise their score, and we are glad that our rebuttal resolved the concerns you had.

---

> > ### Author Response · Authors · 2024-08-14
> >
> > Before the discussion period ends, we were wondering if the reviewer could update their score to reflect the score change? Apologies for the repeated comments.

---

### Official Review · Reviewer_Tak7 · 2024-07-13

**Soundness:** 3
**Presentation:** 3
**Contribution:** 3
**Rating:** 7
**Confidence:** 3

**Summary:**

This paper presents a method to generate multiple sequences from an autoregressive model with a single forward pass. The Superposed Decoding method relies on the approximate linearity of the overall model to additively superpose embeddings for distinct sequences through the model in the same forward pass.

**Strengths:**

The method is shown to have practical benefit over other sampling methods. The idea appears to be novel and is surprising that it works, albeit with an additional n-gram filtering step.

**Weaknesses:**

Understanding this phenomenon better in the architectures tested like LLaMA and Mistral would help the idea in this paper substantially. What is it about the representations that allows them to be processed in superposition and why do these LLMs tend to remain in this condition. Is this impacted by the depth of the network?

There's an important aspect of the kind of linearity that allows the superposition observed here compared to previous work mentioned like [34. 22] cited in this work. The other work showed that representations of concepts are linear, but not that the decoding operations can process superpositions in a linear fashion. Furthermore, the Superposed Decoding method implies that these properties hold for intermediate representations throughout the network.

Also, some relevant work in this direction are worth mentioning:

Elhage, Nelson, et al. "Toy models of superposition." arXiv preprint arXiv:2209.10652 (2022).
Cheung, Brian, et al. "Superposition of many models into one." Advances in neural information processing systems 32 (2019).

**Questions:**

The exploration of the alignment between the component vector and the superposition in Figure 3 is interesting, but could the authors create a baseline to get a better idea of how strong the alignment is. For example, calculating alignment between unrelated sequences would give a better idea of what the expected lower bound of of alignment would be.

For Table 3, again, to get a better idea of the properties of Superposed Decoding, can the authors show what the best perplexity would be (ignoring the fixed compute constraints/comparisons) if one were to generate multiple drafts from Nucleus, Beam/Greedy would be. This is strictly to understand what the difference would be if one were trying to generate the best sequences possible with each type of sampling to see if there's a reduction to that bound when using superposed decoding.

How much does the n-gram "filtering" improve generation quality of the decoding? What are the metrics without this additional step? How much does this step help the other methods like Nucleus/Beam sampling?

**Limitations:**

No societal impact needs to be addressed for this work.

---

> ### Author Rebuttal · Authors · 2024-08-05
>
> We thank the reviewer for the positive review and glad that the reviewer found the idea novel. We appreciate pointing to related work that we forgot to add in the paper initially and will add it. Below, we provide the clarifications requested:
>
> **1. Understanding the phenomenon:** We agree with the reviewer that the success of superposition is a surprising phenomenon and there is a lot to be understood. As pointed out by the reviewer, previous works on understanding superposition (Elhage, Nelson, et al.) and newer updates (Olah et al., July 2024) point to a need for more investigation. We are unsure about exactly what allows representations to be processed while being in superposition, but do observe that this phenomenon happens across various classes of LLMs (Llama, Mistral, OLMo (see below)).
>
> With regards to the impact of network depth, we are unsure what the reviewer means and would appreciate clarification. If they mean the intermediate layers of the language models we currently use, we include an additional figure on linearity by layer in Llama-2-7B, which can be found in Figure 1 of the general response PDF. We will update the paper to contain this. Alternatively, if they mean models of different depth, we run additional experiments on linearity for OLMo-1B, OLMo-7B, Llama-2-13B, and Llama-2-70B. Interestingly, we discover that OLMo-7B is less linear than OLMo-1B. However, Llama-2-13B (40 layers compared to 7B’s 32 layers) is significantly more linear than Llama-2-7B. Linearity further improves on Llama-2-70B (80 layers) from Llama-2-13B. This variance highlights that there is still a lot more about superposition that we can learn.
>
> _Numbers 0-20 in the table represent timesteps._
>
> | Model       | Layers | 0 | 1    | 2    | 3    | 4    | 5    | 6    | 7    | 8    | 9    | 10 | 11   | 12   | 13   | 14   | 15   | 16   | 17   | 18   | 19   | 20      |
> |-------------|-------------|---|------|------|------|------|------|------|------|------|------|-------------|------|------|------|------|------|------|------|------|------|---------|
> | OLMo-1B| 16| 1 | 0.96 | 0.93 | 0.91 | 0.88 | 0.88 | 0.88 | 0.89 | 0.89 | 0.86 | 0.88| 0.87 | 0.88 | 0.87 | 0.87 | 0.85 | 0.86 | 0.85 | 0.85 | 0.85 | 0.87    |
> | OLMo-7B     | 32| 1 | 0.91 | 0.86 | 0.81 | 0.78 | 0.76 | 0.73 | 0.72 | 0.70 | 0.70 | 0.67| 0.65 | 0.65 | 0.60 | 0.58 | 0.59 | 0.59 | 0.60 | 0.57 | 0.59 | 0.58    |
> | Llama-2-7B  | 32| 1 | 0.92 | 0.84 | 0.79 | 0.81 | 0.76 | 0.71 | 0.65 | 0.67 | 0.61 | 0.59| 0.55 | 0.54 | 0.45 | 0.47 | 0.45 | 0.46 | 0.39 | 0.36 | 0.39 | 0.37    |
> | Llama-2-13B | 40| 1 | 0.95 | 0.91 | 0.88 | 0.85 | 0.85 | 0.83 | 0.79 | 0.78 | 0.77 | 0.76| 0.73 | 0.71 | 0.69 | 0.66 | 0.67 | 0.68 | 0.64 | 0.64 | 0.63 | 0.61    |
> | Llama-2-70B | 80| 1 | 0.94 | 0.92 | 0.91 | 0.89 | 0.87 | 0.84 | 0.83 | 0.84 | 0.81 | 0.82| 0.75 | 0.74 | 0.74 | 0.71 | 0.68 | 0.73 | 0.68 | 0.67 | 0.67 | 0.65    |
>
> **2. Observed Linearity:** We agree that our observation of superposition is not the same as other works. Figure 1 in the general response PDF shows that the linearity is more profound in the initial and later layers of the LLM than the middle layers, albeit still much better than random. In addition, while our experiments do suggest that decoding operations can process superpositions linearly, it is important to note that decomposing the superposed representation assumes that the top-k tokens provide a basis for the superposition and still requires further denoising with an n-gram model. We believe there is a lot of interesting future work to be done in this direction.
>
> **3. Linearity analysis with random sequences:** Thanks for the suggestion! Here are results from cosine similarity between random sequences to calibrate what a lower bound would be. Results are averaged over 5000 pairs of random sequences from OpenWebText. We will update our paper with these results as well.
>
> |Timestep|1|2|3 | 4| 5| 6| 7| 8| 9| 10| 11| 12| 13| 14| 15| 16| 17| 18| 19| 20  |
> |-|-|-|-|-|-|-|-|-|-|-|-|-|-|-|-|-|-|-|-|-|
> |**Cosine Similarity**|0.56|0.23|0.17|0.14|0.13|0.12|0.12|0.12|0.12|0.11|0.11|0.12|0.11|0.11|0.11|0.12|0.12|0.12|0.12|0.12|
>
> **4. Best perplexity for other decoding schemes:** Here are the numbers for the average best perplexity for other schemes in case of three drafts. However, we want to add that perplexity is a tricky thing to rely on alone because lower perplexity does not always indicate better coherence (Holtzman et al., ICLR 2020).
> |   | Best Perplexity |
> |--|-|
> | **Superposed Decoding** | 4.63   |
> | **Beam Search**  | 2.87  |
> | **Nucleus Sampling** | 3.47  |
>
> **5. Impact of n-gram filtering:** Thank you for raising this point. Below we show perplexity evaluation without n-gram filtering. At a high-level, one can think of n-gram filtering to be reducing the ambiguity in decomposing a superposed representation and denoising the process. Without this step, while the first draft will be more or less greedy, the other drafts will have significantly worse quality to the detriment of users.
> | Superposed Decoding | Draft 1 | Draft 2 | Draft 3 | Best |
> |-|-|-|-|-|
> | **PPL w/o filtering**| 4.54| 18.33| 18.61| 4.06 |
> | **PPL with filtering**  | 5.03| 7.97| 10.05| 4.63 |
>
> Below, we also show n-gram filtering applied on nucleus sampling. We experiment with several beta values for how heavily n-gram filtering is weighted and evaluate using perplexity. From the experiments, n-gram filtering does not provide significant benefits to nucleus sampling but also does not diminish the performance of nucleus sampling either.
> | Beta (n-gram weight) |0| 0.01 | 0.05 |0.1|0.2|0.4| 0.6| 0.8|
> |-|--|---|--|-|-|-|-|-|
> | **Nucleus Sampling** | 5.17 | 5.18 | 5.26 | 5.16 | 5.24 | 5.16 | 5.15 | 5.11 |
>
> We hope that this rebuttal answers any questions you have and solidifies your positive outlook on the paper. We would love to discuss more if you need any further clarifications.
>
> Ref:
> Olah et al July 2024: https://transformer-circuits.pub/2024/july-update/index.html

---

> > ### Comment · Reviewer_Tak7 · 2024-08-12
> >
> > >  If they mean the intermediate layers of the language models we currently use, we include an additional figure on linearity by layer in Llama-2-7B,
> >
> > Yes, that is what I meant. Thank you for following up on this.
> >
> > > that the top-k tokens provide a basis for the superposition and still requires further denoising with an n-gram model. We believe there is a lot of interesting future work to be done in this direction.
> >
> > If I understand this point correctly, you're referring to the softmax() of the representation when it maps to output tokens is a highly non-linear operation? And the denoising with an n-gram model is meant to repair this issue to some degree?

---

> ### Author Response · Authors · 2024-08-12
>
> Yes, this is exactly correct. Without denoising, it is very difficult to get multiple coherent outputs from the superposed representations.
>
> Another way of thinking of the n-gram models is that they help ground the token distributions in reality to correct for noise in the top-k tokens.

---

### Author Rebuttal · Authors · 2024-08-05

First, we would like to thank all reviewers for their feedback. We would also like to express sincere appreciation to the AC, SAC, and PCs for the time they have put into the current review cycle. We want to reiterate that Superposed Decoding is a novel algorithm leveraging an interesting phenomenon of representational superposition in LLMs towards generating multiple drafts in a single autoregressive inference pass and is complementary to other decoding methods and efficiency improvements.

In the rebuttal PDF, we include a figure on linearity by layer in Llama-2-7B through five timesteps in case it may be of use to the reviewers.

We address the rest of the questions and comments the reviewers had in their respective rebuttals.

---

### Decision · Program_Chairs · 2024-09-25

**Decision:**

Accept (poster)

**Comment:**

The paper presents a novel method for generating multiple coherent text drafts from a single forward pass of an autoregressive model, named Superposed Decoding. The method leverages the approximate linearity of large language models (LLMs) to superpose embeddings of different sequences and decode them in parallel. The paper includes comprehensive experiments demonstrating the method's effectiveness in improving generation perplexity, downstream task performance, and human preference, all while reducing computation time. The concept of Superposed Decoding is novel and has the potential to significantly impact text generation tasks by providing multiple drafts efficiently. The effectiveness of Superposed Decoding relies heavily on the quality of the n-gram models, which could be a point of failure. Most reviewers recognize the novelty of this paper after thorough discussions and I think this paper could be a good contribution to the community.